# Conformational decoupling in acid-sensing ion channels uncovers mechanism and stoichiometry of PcTx1-mediated inhibition

**Stephanie A Heusser[†], Christian B Borg[†], Janne M Colding, Stephan A Pless***

Center for Biopharmaceuticals, Department of Drug Design and Pharmacology, University of Copenhagen, Copenhagen, Denmark

**Abstract** Acid-sensing ion channels (ASICs) are trimeric proton-gated cation channels involved in fast synaptic transmission. Pharmacological inhibition of ASIC1a reduces neurotoxicity and stroke infarct volumes, with the cysteine knot toxin psalmotoxin-1 (PcTx1) being one of the most potent and selective inhibitors. PcTx1 binds at the subunit interface in the extracellular domain (ECD), but the mechanism and conformational consequences of the interaction, as well as the number of toxin molecules required for inhibition, remain unknown. Here, we use voltage-clamp fluorometry and subunit concatenation to decipher the mechanism and stoichiometry of PcTx1 inhibition of ASIC1a. Besides the known inhibitory binding mode, we propose PcTx1 to have at least two additional binding modes that are decoupled from the pore. One of these modes induces a long-lived ECD conformation that reduces the activity of an endogenous neuropeptide. This long-lived conformational state is proton-dependent and can be destabilized by a mutation that decreases PcTx1 sensitivity. Lastly, the use of concatemeric channel constructs reveals that disruption of a single PcTx1 binding site is sufficient to destabilize the toxin-induced conformation, while functional inhibition is not impaired until two or more binding sites are mutated. Together, our work provides insight into the mechanism of PcTx1 inhibition of ASICs and uncovers a prolonged conformational change with possible pharmacological implications.

**\*For correspondence:**
stephan.pless@sund.ku.dk

[†]These authors contributed equally to this work

## Editor's evaluation

This study provides direct evidence that PcTx1, a modulator commonly used to study acid-sensing ion channels, induces a conformational change that persists long after an effect on the channel activity has dissipated. The data support this central claim of the paper and invite future investigation of the precise mechanism. The work is of general interest to those studying ion channel biophysics and pharmacology and is an exemplar of the power of combined functional and fluorescence measurements.

## Introduction

Acid-sensing ion channels (ASICs) are trimeric proton-gated cation channels expressed throughout the central and peripheral nervous system (*Gründer, 2020*). The extracellular domains (ECDs) of ASICs sense pH changes, allowing cations to permeate through the channel pore comprised by the transmembrane domain (TMD) (*Figure 1A*). Research over the last two decades has shown that this pH-activated current underlies various physiological processes, such as fast synaptic transmission (*Du et al., 2014*) and learning and memory (*Wemmie et al., 2002*). Due to recent reports of peptide-induced

**Figure 1.** Psalmotoxin-1 (PcTx1) induces a distinct long-lived conformational state in the acid-sensing ion channel 1a (ASIC1a) extracellular domain (ECD). (**A**) Structural overview (PDB ID 4FZ0) of chicken ASIC1 with positions V80 and K105, which were substituted for cysteine and used for channel labeling, highlighted in red. PcTx1 (teal) binds to the subunit interfaces. (**B**) Representative two-electrode voltage-clamp (TEVC) traces recorded from *X. laevis* oocytes of WT ASIC1a showing pH sensitivity of activation in the absence (upper panel) and presence (lower panel) of 30 nM PcTx1, added in the resting solution (pH 7.9). Scale bars are 4 μA (vertical) and 60 s (horizontal). (**C**) Same as in (**B**) but for steady-state desensitization (SSD). PcTx1 was applied to solutions of decreasing pH in between application of activating pH 5.6 solution. Scale bars are 4 μA (vertical) and 60 s (horizontal). (**D**) Concentration–response relationship of WT ASIC1a activation and SSD in the absence and presence of 30 nM PcTx1 retrieved form experiments shown in (**B**) and (**C**) (n = 6–18). (**E**) Representative traces of voltage-clamp fluorometry (VCF) recordings of K105C* with the current in black and the fluorescence in red. PcTx1 (300 nM) was washed off for 3 min using pH 7.4 (left) or pH 8.4 (right). Scale bars are 60 s (black horizontal), 10 μA (black vertical), and 10% (red vertical). (**F**) Quantitative analysis of the fluorescence signal at the end of the 3 min washout protocols shown in (**E**) relative to the fluorescence observed upon PcTx1 application. (**G**) Representative trace of a VCF recording of V80C* equivalent to the ones shown in (**E**). Scale bars are 60s (black horizontal), 10 μA (black vertical), and 10% (red vertical).(**H**) Same as in (**F**) but for V80C*F350L. Data in (**D**), (**F**), and (**H**) are presented as mean ± 95 CI.

The online version of this article includes the following source data and figure supplement(s) for figure 1:

**Source data 1.** TEVC data from mASIC1a WT of activation and SSD with and without PcTx1, as shown in *Figure 1B-D*.

**Source data 2.** VCF data from K105C* and V80C* of different PcTx1 washout protocols, as shown in *Figure 1E-H* and *Figure 1—figure supplement 2*.

**Figure supplement 1.** Voltage-clamp fluorometry (VCF) data of pH activation.

**Figure supplement 1—source data 1.** VCF data from mASIC1a K105C* and K105C*F350L of the pH dependent changes in fluorescence and SSD, as shown in *Figure 1—figure supplement 1*.

**Figure supplement 1—source data 2.** VCF data from mASIC1a V80C* and V80C*F350L of the pH dependent changes in fluorescence and SSD, as shown in *Figure 1—figure supplement 1*.

**Figure supplement 2.** Voltage-clamp fluorometry (VCF) data of psalmotoxin-1 (PcTx1) washout.

ASIC1a inhibition exhibiting neuroprotective effects, these trimeric channels have emerged as a potential drug target in neurological diseases, including protection from acidosis-induced neurotoxicity and reduction of infarct volume in mouse models of ischemic stroke (*Xiong et al., 2004*; *Wemmie et al., 2013*; *Chassagnon et al., 2017*; *Qiang et al., 2018*; *Gründer, 2020*; *Heusser and Pless, 2021*).

ASICs exhibit a rich pharmacology with complex molecular details. Modulators can affect both ASIC activation and steady-state desensitization (SSD)—a state reached at sub-activating proton concentrations that reduces subsequent activation—but also metabotropic downstream signaling (*Kellenberger and Schild, 2015*; *Wang et al., 2015*; *Wang et al., 2020*). Venom-derived peptides are currently the most selective and potent modulators of ASIC function and therefore serve as valuable tools to gain insights into the mechanisms of ASIC modulation (*Cristofori-Armstrong and Rash, 2017*). Psalmotoxin-1 (PcTx1) is a 40-residue cysteine knot peptide toxin extracted from tarantula venom (*Escoubas et al., 2000*). It acts as a potent gating modifier on most ASIC1a-containing channels (*Chen et al., 2005*; *Chen et al., 2006*) and shifts activation and SSD curves to more alkaline values, thereby effectively inhibiting the channel in the low nanomolar range at neutral pH (*Cristofori-Armstrong and Rash, 2017*). Structural studies have shown that PcTx1 binds to the subunit interfaces of cASIC1, forming contact points with the primary and complementary side of the acidic pocket (*Dawson et al., 2012*; *Baconguis et al., 2014*; *Figure 1A*). An aromatic side chain located at the entrance of the acidic pocket, F352 in human ASIC1a (F350 in mouse ASIC1a), plays an important role in PcTx1 activity because mutations to either alanine or leucine (or ASIC isoforms that naturally carry these residues at this position) display significantly attenuated PcTx1 sensitivity (*Sherwood et al., 2009*; *Sherwood et al., 2011*; *Saez et al., 2015*; *Joeres et al., 2016*; *Cristofori-Armstrong et al., 2019*). Additional species- and state-dependent factors greatly influence the functional outcome of PcTx1 modulation (*Sherwood et al., 2011*; *Cristofori-Armstrong et al., 2019*), but fundamental mechanistic aspects of this interaction remain unresolved. For example, it remains unclear how many toxin molecules are required to obtain functional inhibition, despite the availability of structural information on the PcTx1-ASIC1 complex (*Baconguis and Gouaux, 2012*; *Dawson et al., 2012*). Further, little is known about the conformational consequences of the interaction in different functional states and under varying pH conditions. This is relevant since long-lived conformational changes could affect the activity of endogenous ASIC1a modulators.

Here, we show that PcTx1 adopts at least three distinct binding modes, of which one induces a long-lived conformational state in the ASIC1a ECD that alters the effects of the endogenous neuropeptide big dynorphin (BigDyn). Tracking the ECD conformational state further uncovers an unexpected decoupling from the state of the pore that has profound implications for the stoichiometry of the ASIC1a-PcTx1 interaction: while disrupting a single PcTx1 binding site in an ASIC1a trimer is sufficient to destabilize the long-lived PcTx1-induced ECD conformational state, functional inhibition of the channel is only affected when two or more PcTx1 binding sites are mutated. Collectively, we illustrate how a combination of voltage-clamp fluorometry (VCF) and subunit engineering provides detailed mechanistic insights on a potentially therapeutically relevant channel-toxin interaction and demonstrate how these findings affect fundamental pharmacological properties of the channel.

## Results

### PcTx1 induces a long-lived ECD conformation in ASIC1a

First, we sought to establish the functional consequences of PcTx1 binding to mASIC1a. In line with previous findings, 30 nM of PcTx1 led to potent alkaline shifts in both the activation curve ($\Delta pH_{50\_Act}$ > 0.5 pH units at pH 7.9; *Figure 1B and D*, *Supplementary file 1a*) and SSD of ASIC1a WT expressed in *Xenopus laevis* oocytes ($\Delta pH_{50\_SSD}$ ≈ 0.4 pH units; *Figure 1C and D*, *Supplementary file 1a*; *Escoubas et al., 2000*; *Chen et al., 2005*). This electrophysiological characterization provides a detailed description of how PcTx1 affects channel function, as assessed by the state of the pore. By contrast, it is unable to capture the extent and time course of possible conformational consequences caused by the toxin-channel interaction in the ECD of ASIC1a. It therefore remained unclear if potential toxin-induced conformational changes extend across the entire ECD and how long they persist. To this end, we introduced a cysteine residue into the mASIC1a ECD at amino acid position 80 or 105 (*Figure 1A*) as labeling of these positions with the environmentally sensitive Alexa Fluor 488 dye reliably reports on conformational changes related to channel gating and modulation by peptides (*Bonifacio et al., 2014*; *Borg et al., 2020*). This allowed us to monitor changes in the fluorescence signal upon exposure to different proton or toxin concentrations and thus provide a proxy for ECD conformational changes at two spatially distant positions. For Alexa Fluor 488-labeled K105C mASIC1a (K105C*), application of pH 5.5 resulted in an inward current and a simultaneous upward deflection of

the fluorescence signal (*Figure 1E*). The latter typically displayed a transient peak, which likely reflects a short-lived open state, followed by a sustained plateau with a pH dependence that was similar to the $pH_{50\_SSD}$, and thus likely reveals a transition related to a desensitized state (*Figure 1—figure supplement 1*, *Supplementary file 1b*; note that pH changes between 7.4 and 9.0 did not lead to any fluorescence changes *Borg et al., 2020*). Application of 300 nM PcTx1 at pH 7.4 resulted in slow upward deflection of the fluorescence (*Figure 1E*, *Supplementary file 1c*), reminiscent of application of pH 7.0, which led to a similar functional inhibition of channel currents (*Figure 1—figure supplement 1A*). However, and in stark contrast to the pH 7.0 application, the PcTx1-induced fluorescence signal displayed only a modest decrease after 3 min washout with pH 7.4. Post-PcTx1 exposure to pH 8.4 led to an immediate return of fluorescence to baseline level; yet, the fluorescence increased as soon as the pH was switched back to 7.4, suggesting the presence of a conformational change that is sustained at pH 7.4 (*Figure 1E*, left panel). We observed similar effects with two other pH application protocols after PcTx1 exposure (*Figure 1E*, right panel, *Figure 1F*, *Figure 1—figure supplement 2A and B*, *Supplementary file 1d*).

To rule out a direct interaction between the fluorophore and PcTx1 or the possibility of a local conformational change, we also labeled the channel in position V80C of the palm domain (V80C*, *Figure 1A*) and characterized the pH and PcTx1 responses. Here, the direction of the fluorescence signals upon activation, desensitization, and application and washout of 300 nM PcTx1 at 0.3 pH units more alkaline than the $pH_{50\_fluorescence}$ of V80C* were inverted (*Figure 1G*), but resulted in qualitatively similar effects as observed for K105C* (*Figure 1E–H*, *Figure 1—figure supplement 2C and D*, *Supplementary file 1c and d*). In light of the data obtained from K105C* and V80C*, we considered it implausible that the fluorescence changes are the consequence of a direct interaction between the dye and the peptide, but rather result from a global conformational change of the ECD after PcTx1 binding. We conclude that PcTx1 binding induces a long-lived conformational change and that this conformational alteration is not necessarily coupled to the channel pore.

## PcTx1 has three distinct binding modes, and pre-exposure can alter channel pharmacology

Next, we investigated the possibility that PcTx1 may have multiple distinct binding modes, as well as potential functional and pharmacological implications of the PcTx1-induced conformational states.

For this purpose, we first sought to define the PcTx1-induced inhibitory state in more detail. This tightly bound PcTx1 state leads to the characteristic functional change of the pore (i.e., current inhibition) *and* to a change in fluorescence (*Figure 1E and G*, *Figure 1—figure supplement 2A*, *Figure 2A*, *Figure 2—figure supplement 1A and C*). We thus term it the 'Global' state. Due to the gating-modifying properties of PcTx1 on activation and SSD, the functional consequences of this 'Global' state are highly pH dependent. In the presence of PcTx1 and at neutral pH, subsequent channel activation is temporarily inhibited (*Figure 1E and G*, *Figure 1—figure supplement 2A*, *Figure 2A*, *Figure 2—figure supplement 1A*), while exposure at high pH shows increased activation (*Figure 2B*, middle, *Figure 2—figure supplement 1C*).

We found that once the channel is in this 'Global' state, repeated exposure to pH 5.5 led to a gradual recovery of the current amplitude, while the fluorescence signal remained effectively unchanged (*Figure 2A*). Functional recovery from inhibition was dependent on washout time, but not activation (*Figure 2—figure supplement 1A and B*, *Supplementary file 1e*), in line with previous observations of tight PcTx1 binding to the open state (*Chen et al., 2006*). This finding emphasized a clear decoupling between the conformational state of the ECD and the functional inhibitory consequences at the level of the pore: while the pore slowly recovered from its inhibited state and regained its ability to conduct ions, the ECD conformation remained essentially unaltered, even after minutes of washout. We therefore suggest the presence of an 'ECD$_{only}$' state, which is present following PcTx1 exposure and exists at neutral/low pH in the absence of PcTx1 in the solution. It causes long-lasting conformational alterations in the ECD but has no inhibitory effects on the pore (*Figure 2A*).

Lastly, we sought to address the possibility of additional PcTx1-induced states beyond the 'Global' and the 'ECD$_{only}$' state. To this end, we conducted a series of experiments in which we applied PcTx1 at pH 8.0, where PcTx1 neither activates the channel nor induces a significant increase in fluorescence. (i) If immediately after PcTx1 application (at pH 8.0) the channel is activated in the absence of PcTx1 and then washed with pH 7.4, the channel can enter into the long-lived 'ECD$_{only}$' state (*Figure 2B*, left

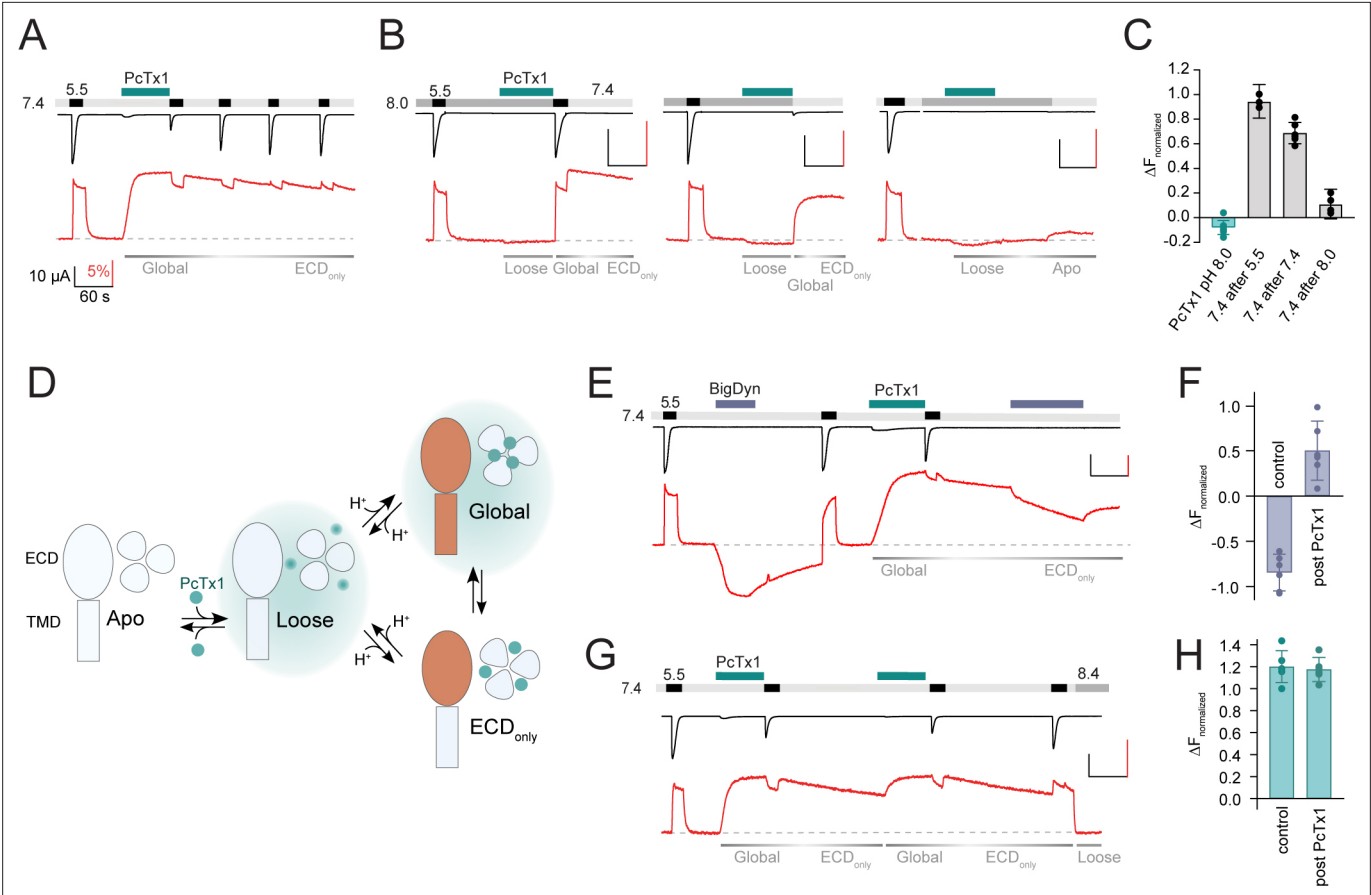

**Figure 2.** Psalmotoxin-1 (PcTx1) has pH-dependent binding modes and alters acid-sensing ion channel 1a (ASIC1a) pharmacology. (**A**) Voltage-clamp fluorometry (VCF) trace of K105C* showing the introduction of the 'Global' inhibitory binding mode upon application of 300 nM PcTx1 at pH 7.4. During washout and repeated activation, the channel readily returns to a functional *apo* state (current, black trace) while the fluorescence change induced by PcTx1 is persistent over multiple ASIC1a activations at pH 5.5 (fluorescence, red trace), characteristic for the 'ECD_only' state. (**B**) VCF traces highlighting the fluorescence changes associated with application of PcTx1 at pH 8.0 with subsequent application of pH 5.5 (left), pH 7.4 (middle), and pH 8.0 (right). Respective PcTx1 binding modes are indicated below the traces. (**C**) Quantitative comparison of the fluorescence signal 60 s into the pH 7.4 application at the end of the experiments shown in (**B**) normalized to the fluorescence change induced by pH 5.5 application. (**D**) Schematic representation of the different pH-dependent binding modes of PcTx1: A 'Loose' closed state at high pH, a 'Global' state that exists at neutral/low pH that leads to conformational rearrangements in the extracellular domain (ECD) and the pore (indicated in orange), and an 'ECD_only' state in which the conformational rearrangements are only found in the ECD and that exists at neutral/low pH even when PcTx1 is absent in the extracellular solution. Teal background shading in the 'Loose' and 'Global' indicates the presence of PcTx1 in the extracellular solution (although not mandatory, see text for details). (**E**) VCF trace of K105C* exposed to pH 5.5, followed by a 60 s big dynorphin (BigDyn) (1 μM) application (purple bar), with subsequent washout and activation. BigDyn is reapplied after the 'ECD_only' state has been evoked through PcTx1 (300 nM) application, this time resulting in a smaller decrease in the fluorescence signal. (**F**) Quantitative comparison of the fluorescence change induced by a 60 s BigDyn application to the *apo* (control) and to the PcTx1-induced 'ECD_only' state (post PcTx1), normalized to the signal induced by pH 5.5. (**G**) VCF trace of K105C* where 300 nM PcTx1 is applied to the 'ECD_only' state. (**H**) Quantitative analysis of the protocol shown in (**G**) comparing the fluorescence change induced by PcTx1 to the *apo* state at 7.4 (control) with the PcTx1 application to the 'ECD_only' state. All scale bars are 60 s (black horizontal), 10 μA (black vertical), and 5% (red vertical). Data in (**C**), (**F**), and (**H**) are presented as mean ± 95 CI.

The online version of this article includes the following source data and figure supplement(s) for figure 2:

**Source data 1.** VCF data from mASIC1a K105C* of single and multiple activations during PcTx1 washout, as shown in *Figure 2A* and *Figure 2—figure supplement 1A, B*.

**Source data 2.** VCF data from mASIC1a K105C* of PcTx1 application at pH 8.0 followed by different washout protocols, as seen in *Figure 2B, C*.

**Source data 3.** VCF data of mASIC1a K105C* of BigDyn and PcTx1 application, as seen in *Figure 2E-H* and *Figure 2—figure supplement 1D, E*.

**Figure supplement 1.** Voltage-clamp fluorometry (VCF) data of psalmotoxin-1 (PcTx1) modulation and pharmacology.

panel, *Figure 2C*, *Supplementary file 1f*). (ii) Similarly, the 'ECD$_{only}$' state can be reached by administering PcTx1 during a pH 8.0-induced closed state before exposing the now highly proton-sensitive channel to pH 7.4 or 7.0, which results in temporary activation and SSD (*Figure 2B*, middle panel, *Figure 2C*, *Figure 2—figure supplement 1C*, *Supplementary file 1f*). Thus, while exposure of PcTx1 at pH 8.0 has no effect on current and fluorescence, it primes the channel into a state where subsequent activation or desensitization can elicit a long-lived PcTx1-induced conformational state that is decoupled from the pore. (iii) But when a pH 8.0-induced closed state is maintained before, during, and after PcTx1 application, pH 7.4 is no longer able to elicit a conformational change (*Figure 2B*, right panel, *Figure 2C*, *Supplementary file 1f*). This indicates that PcTx1 binding to the closed state is readily reversible at high pH. Based on the findings from these three experiments, we infer the presence of a third state that we term 'Loose': a loosely bound high pH state that is not associated with ECD conformational changes that can be detected by VCF, nor with changes to the functional state of the pore. The 'Loose' state can exist in the presence (*Figure 2B*) or after the exposure to PcTx1 (*Figure 1E and G*, *Figure 1—figure supplement 2A and C*, *Figure 2—figure supplement 1C*) and prolonged exposure of this 'Loose' state to high pH can wash off PcTx1 (*Figure 2B*, right). A graphical summary of the proposed three states ('Loose,' 'Global,' and 'ECD$_{only}$') is provided in *Figure 2D*.

If correct, the 'ECD$_{only}$' state would be expected to affect the activity of other ECD-targeting ASIC1a modulators. To test this possibility, we turned to BigDyn, a neuropeptide that competes with PcTx1 for the binding site and induces a distinct closed state in ASIC1a (*Sherwood and Askwith, 2009*; *Borg et al., 2020*; *Leisle et al., 2021*). The latter is indicated by a robust and distinct downward deflection of the K105C* fluorescence signal (*Figure 2E*). Strikingly, we found that when BigDyn (1 μM) is applied to the PcTx1-induced 'ECD$_{only}$' state, the neuropeptide has a much less pronounced effect on the fluorescence signal than without PcTx1 pre-exposure (*Figure 2E and F*, *Supplementary file 1g*). To ensure that this effect is indeed specific to PcTx1, we conducted a control experiment in which we first applied BigDyn and then switched to PcTx1 (*Figure 2—figure supplement 1D*). In this case, the PcTx1-elicited fluorescence response post BigDyn exposure was unchanged compared to that observed in control cells (*Figure 2—figure supplement 1E*, *Supplementary file 1g*). Conversely, reapplication of PcTx1 to the 'ECD$_{only}$' state evoked a similar inhibitory and fluorescence response as the initial application (*Figure 2G and H*, *Supplementary file 1g*), indicating that a transition from the 'ECD$_{only}$' to the 'Global' state is possible. In summary, these findings support the notion that the 'ECD$_{only}$' state, although without immediate inhibitory consequences for the channel pore, does have direct implications for binding of other ECD-targeting compounds.

## F350L diminishes PcTx1 sensitivity and stability of toxin-induced conformation

Next, we wanted to establish how PcTx1 sensitivity and binding modes change in the F350L mutation that renders the channel virtually insensitive to the toxin (*Figure 3A*; *Sherwood et al., 2009*; *Saez et al., 2015*). We thus compared the PcTx1 sensitivity of the F350L mutant to that of WT mASIC1a at a pH that would neither activate nor steady-state desensitize the channels in the absence of the toxin (pH 7.9 for WT; pH 7.6 for F350L). Consistent with previous findings (*Sherwood et al., 2009*), introducing the F350L mutation reduced the pH sensitivity of activation and SSD, and the presence of 30 nM PcTx1 only led to a slight shift in pH$_{50\_SSD}$, with no discernible effect on channel activation (*Figure 3B and C*, *Supplementary file 1a*). The inhibitory effect of PcTx1 under physiological pH (7.4) was still concentration dependent but dramatically decreased in the F350L variant compared to WT (IC$_{50}$ ~ 1 μM for F350L compared to ~1 nM for WT; *Figure 3D and E*, *Supplementary file 1h*). Introduction of the F350L mutation therefore strongly reduced the gating-modifying properties of PcTx1 and diminished its inhibitory effect by three orders of magnitude.

In contrast to the K105C* variant, we found that application of 300 nM PcTx1 to K105C*F350L at pH 7.4 did not elicit any fluorescence changes (*Figure 3F*, *Supplementary file 1c*). Notably, the fluorescence response curve of the K105C*F350L variant is about 0.1 pH units shifted towards more acidic pH compared to K105C* (*Figure 1—figure supplement 1B*, *Supplementary file 1b*). Taking this change into account, we applied PcTx1 at pH 7.3. Under these conditions, PcTx1 induced a similar slow-onset fluorescence signal as seen for K105C*, albeit slightly less pronounced, while the functional insensitivity of the F350L variant to PcTx1 remained (*Figure 3G*, *Supplementary file 1c*), indicating that the channel can still enter the 'ECD$_{only}$' state while the 'Global' binding mode is

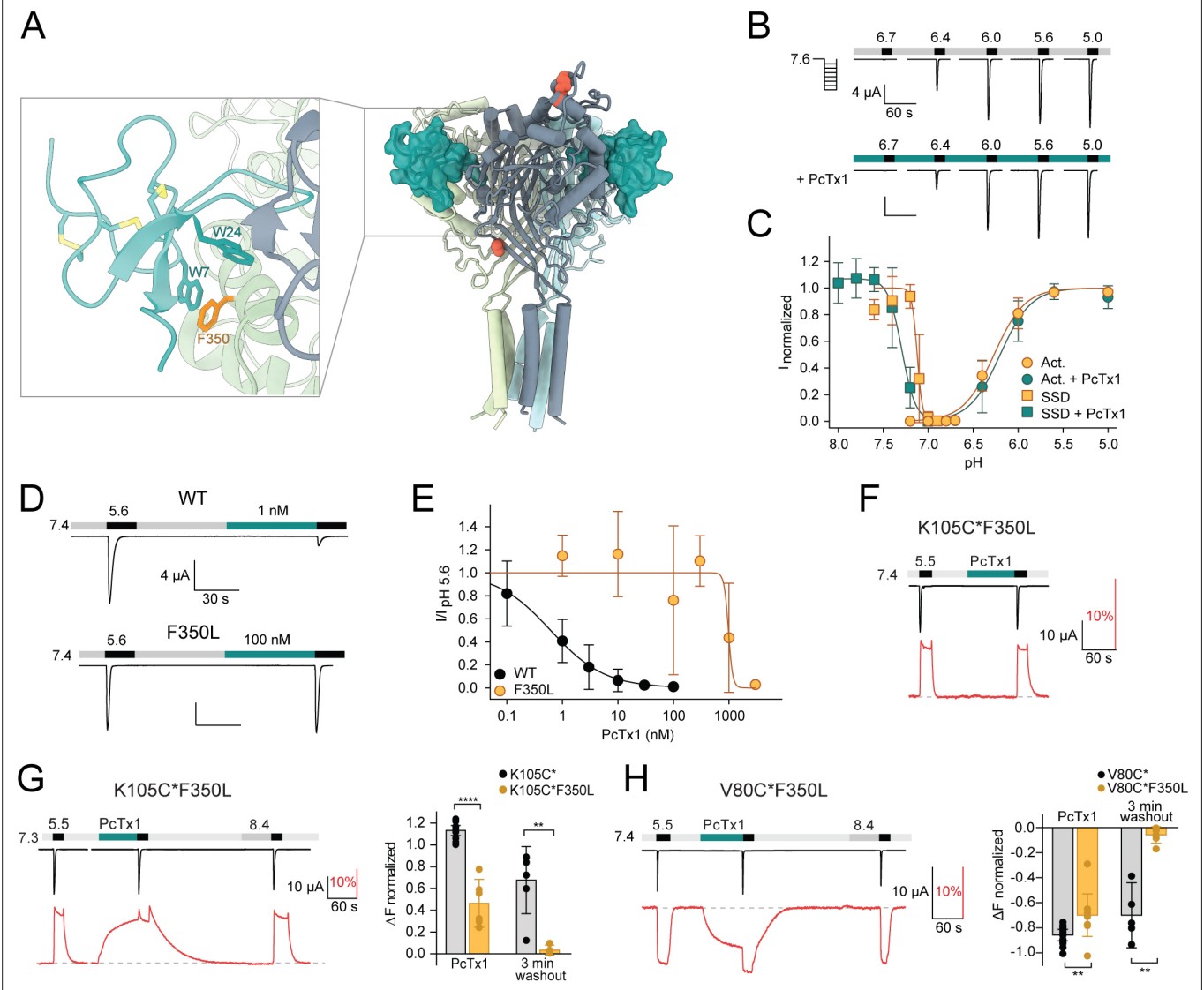

**Figure 3.** The F350L mutation diminishes psalmotoxin-1 (PcTx1) sensitivity by destabilizing the toxin-induced conformation. (**A**) Model of the co-crystal structure of cASIC1a and PcTx1 (teal) binding to the extracellular domain (PDBID 4FZO). Inset shows a closeup of the interaction site at the acidic pocket, including acid-sensing ion channel 1a (ASIC1a) residue F350 (orange). (**B**) Representative two-electrode voltage-clamp (TEVC) trace showing mASIC1a F350L pH activation in the absence (top) and presence (bottom) of 30 nM PcTx1. Scale bars are 4 µA (vertical) and 60 s (horizontal). (**C**) Activation and steady-state desensitization (SSD) curve of F350L mASIC1a without (orange) and with (teal) 30 nM PcTx1 (n = 5–12). (**D**) TEVC traces showing the effect of 1 nM PcTx1 on mASIC1a WT (top) and 100 nM PcTx1 on F350L (bottom) applied at pH 7.4. Scale bars are 4 µA (vertical) and 30 s (horizontal). (**E**) PcTx1 concentration–response curve at pH 7.4 using the protocol shown in (**D**) (WT: n = 9–14, F350L: n = 4–7). (**F**) Representative voltage-clamp fluorometry (VCF) trace of the K105C*F350L mutant showing application of 300 nM PcTx1 at pH 7.4. (**G**) Left: representative VCF trace of the K105C*F350L mutant showing application of 300 nM PcTx1 at pH 7.3. Right: comparison of the fluorescence change upon PcTx1 application and after a 3 min washout between K105C* and K150C*F350L. (**H**) Left: representative trace of a VCF recording of V80C*F350L equivalent to the one shown in (**G**). Right: same analysis as in (**G**) but compared between V80C* and V80C*F350L. Data in (**C**), (**E**), (**G**), and (**H**) are presented as mean ± 95 CI, unpaired Mann–Whitney test, **p<0.005, ****p<0.0001.

The online version of this article includes the following source data for figure 3:

**Source data 1.** TEVC data from mASIC1a F350L of activation and SSD with and without PcTx1, as shown in *Figure 3B, C*.

**Source data 2.** TEVC data from mASIC1a WT and F350L of PcTx1 concentration-response curve, as shown in *Figure 3D, E*.

**Source data 3.** VCF data from mASIC1a K105C*(F350L) and V80C*(F350L) of application and washout of PcTx1, as shown in *Figure 3F-G*.

unfavorable. A striking difference was apparent in the washout: unlike in K105C*, the fluorescence signal of K105C*F350L readily returned to a level indicative of a closed *apo* state in the first 1–2 min of pH 7.4 washout, suggesting that the longevity of the 'ECD$_{only}$' state is destabilized (***Figure 3G***, ***Supplementary file 1d***). Again, these observations were mirrored by the V80C*F350L variant (***Figure 3H***, ***Supplementary file 1c and d***), thereby emphasizing the robust and ECD-wide nature of the observed conformational changes. Together, these data suggest that the functional PcTx1 insensitivity of the F350L variant primarily originates from a destabilization of the PcTx1-induced conformational states ('Global' and 'ECD$_{only}$') rather than solely from impaired binding.

## Contribution of individual F350L subunits to PcTx1 inhibition

The drastic reduction in PcTx1 sensitivity and notable destabilization of the PcTx1-induced 'ECD$_{only}$' state through the F350L mutation provided a unique opportunity to determine the stoichiometric requirements for functional PcTx1 modulation, as well as the contribution of individual F350L-containing ASIC1a subunits to PcTx1 insensitivity. Conventional approaches are inapt to answer these fundamental questions as it is technically challenging to introduce targeted mutation in a single subunit within a homotrimer, and co-expression of non-concatenated WT and mutant subunits can lead to undesired, heterogeneous subunit assemblies precluding accurate interpretation of functional data. The use of concatemeric constructs helps to overcome these limitations. By fusing cDNAs of subunits of interest in the desired order, this approach predetermines the subunit composition and assembly of channels, allowing us to directly address questions such as the number of intact PcTx1 binding sites needed for functional inhibition. Importantly, the trimeric assembly of ASICs has proven to be highly amenable to this approach (***Joeres et al., 2016***; ***Lynagh et al., 2017***; ***Wu et al., 2019***).

We therefore designed concatemeric ASIC1a constructs of triple WT subunits (WT/WT/WT) and constructs carrying the F350L substitution in one, two, or all three subunits (F350L/F350L/F350L) using a previously validated construct design (***Figure 4A***, ***Figure 4—figure supplement 1***; ***Lynagh et al., 2017***). The pH dependence of the WT/WT/WT and the F350L/F350L/F350L concatemer constructs showed similar pH sensitivities compared to homotrimeric WT and F350L, respectively (***Figure 4—figure supplement 2A***, ***Supplementary file 1i***). The number of F350L subunits determined the pH dependence of concatemeric ASIC1a constructs but was not dependent on the relative positioning of the mutation within the concatemeric ASIC1a construct (***Figure 4B–E***, ***Figure 4—figure supplement 2B***, ***Supplementary file 1i***). These results provided a robust foundation to assess the effect of one, two, or three F350L-bearing subunits on PcTx1 inhibition.

For the WT/WT/WT concatemer, pH 5.6-induced currents were inhibited by PcTx1 with an apparent IC$_{50}$ of around 6 nM at pH 7.4 compared to an IC$_{50}$ of 0.6 nM for WT ASIC1a (***Figure 3E***, ***Figure 4F and G***, ***Supplementary file 1h***). For the F350L/F350L/F350L concatemer, PcTx1 inhibited pH 5.6-induced currents with an apparent IC$_{50}$ of around 800 nM, similar to the trimeric F350L ASIC1a mutant (***Figure 3E***, ***Figure 4F and G***, ***Supplementary file 1h***). This meant that despite the lower PcTx1 sensitivity of the WT/WT/WT concatenated construct the substantial decrease in PcTx1 sensitivity caused by the F350L mutation was maintained in the F350L/F350L/F350L construct. For concatenated constructs containing two F350L-bearing subunits (WT/F350L/F350L), pH 5.6-induced currents were inhibited by PcTx1 with an apparent IC$_{50}$ around 100 nM (***Figure 4F and G***, ***Figure 4—figure supplement 2C***, ***Supplementary file 1h***). This represented a 17-fold increase in apparent IC$_{50}$ compared to the WT/WT/WT concatemer and an 8-fold decrease compared to the F350L/F350L/F350L concatemer. Remarkably, the concatemer with only a single F350L-containing subunit (WT/F350L/WT) closely matched the PcTx1 inhibition profile observed for the WT/WT/WT concatemer (***Figure 4F and G***, ***Figure 4—figure supplement 2C***, ***Supplementary file 1h***).

Taken together, we demonstrate that application of PcTx1 at pH 7.4 maintains the WT-like inhibitory effect on pH 5.6 activation even if one out of three subunits contains the F350L mutation. We conclude that two WT subunits per mASIC1a trimer are sufficient to observe WT-like functional inhibition by PcTx1.

## PcTx1-induced conformational changes in concatemeric channels

We next wanted to test if a stepwise introduction of F350L-bearing subunits would result in a difference in the stability of the PcTx1-induced 'ECD$_{only}$' state observed in VCF experiments. To this end, we introduced the K105C mutation in all three subunits for Alexa Fluor 488 labeling, while increasing the

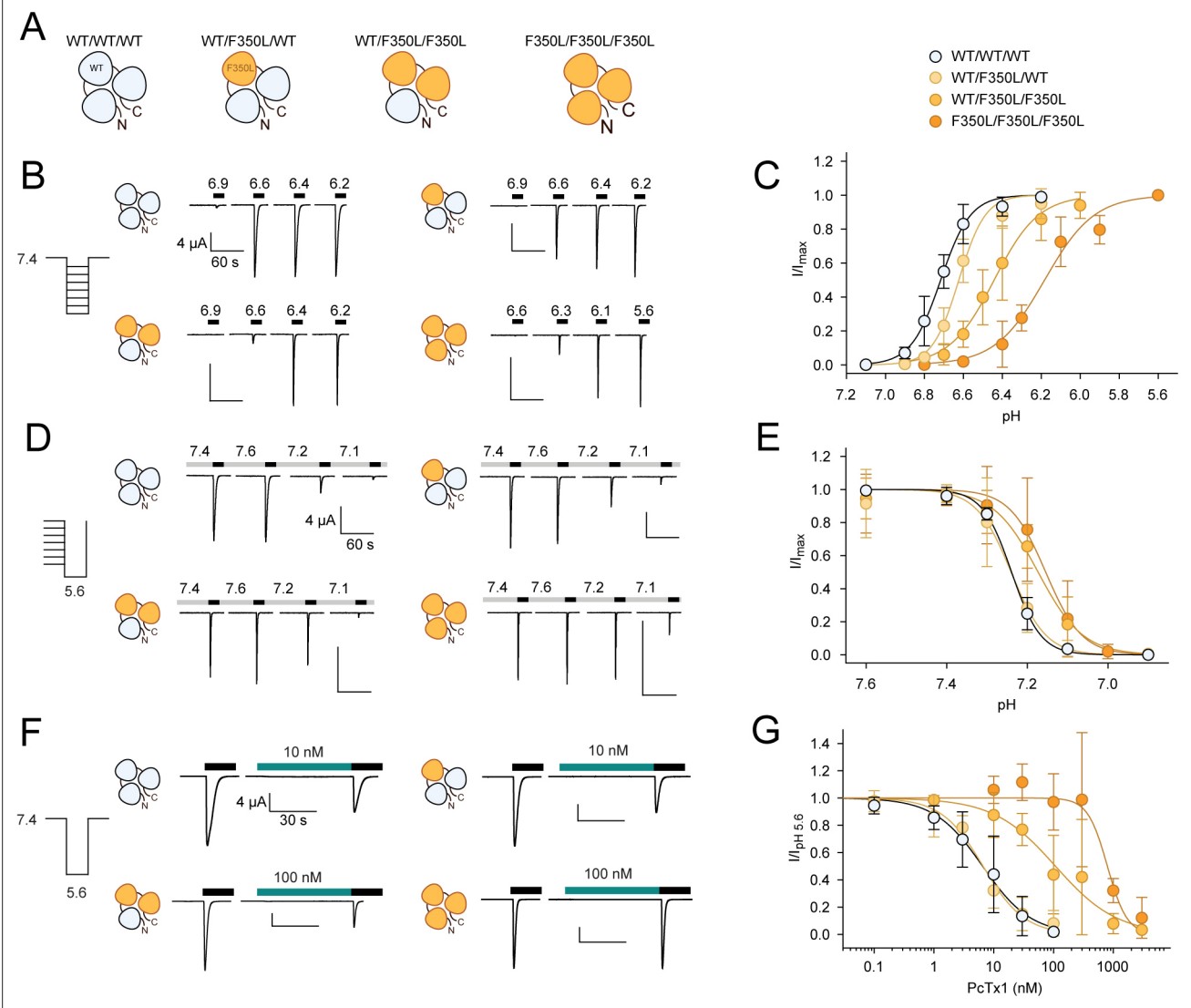

**Figure 4.** In concatemeric constructs, two WT subunits are sufficient for WT-like psalmotoxin-1 (PcTx1) inhibition. (**A**) Schematic overview of the concatemeric constructs containing the F350L mutation (orange) in none, one, two, or all three subunits. (**B**) Representative two-electrode voltage-clamp (TEVC) trace of activation (**C**) Activation curve from recordings shown in (**B**) for the four different concatemeric constructs (n = 7–13). (**D**) Representative TEVC trace of steady-state desensitization (SSD). (**E**) SSD profiles from recordings shown in (**D**) (n = 4–11). (**F**) Representative TEVC trace of concentration-dependent PcTx1 inhibition at pH 7.4. (**G**) PcTx1 concentration–response curves from data shown in (**F**) (n = 4–11). Scale bars are 4 µA (vertical) and 60 s (horizontal) for (B, D) and 30 s for (F). Data points in (**C, E and G**) represent mean ± 95CI.

The online version of this article includes the following source data and figure supplement(s) for figure 4:

**Source data 1.** TEVC data from concatemeric mASIC1a of pH-dependent activation, as shown in *Figure 4B, C* and *Figure 4—figure supplement 2B*.

**Source data 2.** TEVC data from concatemeric mASIC1a of SSD, as shown in *Figure 4D, E*.

**Source data 3.** TEVC data from concatemeric mASIC1a of PcTx1 concentration–response curve, as shown in *Figure 4F, G* and *Figure 4—figure supplement 2C*.

**Figure supplement 1.** Concatemer design and validation.

**Figure supplement 2.** Sensitivity of concatemers to pH and psalmotoxin-1 (PcTx1).

number of F350L-containing subunits from one to two and three, respectively. We refer to subunits with WT background but labeled at K105C as WT*, while subunits labeled at K105C *and* containing the F350L mutation are designated F350L*. First, we established the pH dependence of the fluorescence signal. Overall, the $pH_{50\_fluorescence}$ and the $pH_{50\_SSD}$ matched in all constructs and were in a similar range as the non-concatenated channels (*Figure 5—figure supplement 1*, *Supplementary file 1b*).

PcTx1 (300 nM) was applied 0.3 pH units above the measured $pH_{50\_fluorescence}$. In the WT*/WT*/WT* concatemer, PcTx1 functionally inhibited channel activation and led to the characteristic increase in fluorescence maintained even after removal of PcTx1 from the buffer, and this was comparable to what we observed in the K105C* trimer (*Figure 1E*, *Figure 5A, D, and E*, *Supplementary file 1c and d*).

In WT*/F350L*/WT*concatemers, PcTx1 still had an inhibitory effect on activation, in line with the results from nonlabeled constructs, and produced a robust fluorescence response (*Figure 5B and D*, *Supplementary file 1c*). In this construct, a marked gradual decrease of the fluorescence signal during the 3 min washout at pH 7.4 was noticeable (*Figure 5B and E*, *Supplementary file 1d*). In constructs with two F350L-containing subunits, the response of the fluorescence signal to PcTx1 was further reduced, and the fluorescence signal returned to baseline levels (indicative of a closed *apo* state) in the first 1–2 min of the pH 7.4 washout (*Figure 5C–E*, *Supplementary file 1c and d*), reminiscent of the PcTx1 effect on the non-concatemeric K105C*/F350L channel (*Figure 3G*). No reliable changes in fluorescence could be detected for the F350L*/F350L*/F350L*, likely due to low expression.

In summary, our VCF experiments demonstrate that in contrast to the functional modulation the presence of a single F350L-containing subunit is sufficient to destabilize the 'ECD_only' state introduced by PcTx1, further emphasizing that PcTx1 can elicit distinct responses in the ECD and the pore of ASIC1a (*Figure 6*).

## Discussion

PcTx1 is a widely used ASIC1a modulator in in vitro and in vivo studies, yet the molecular underpinnings of binding, resulting conformational changes, and stoichiometric requirements remained enigmatic. Here, we employ VCF and concatemeric ASIC1a constructs that carry a virtually PcTx1-insensitive mutation in one, two, or all three subunits to establish the stoichiometry of the ASIC1a-PcTx1 interaction. We establish three binding modes for PcTx1 and observe that at least one of them induces a 'ECD_only' conformational change, which leaves the state of the pore unaffected but impacts binding of an endogenous peptide modulator. Collectively, our work provides mechanistic insight into the ASIC1a-PcTx1 interaction and emphasizes the potential of protein engineering approaches for detailed studies of protein−peptide interactions.

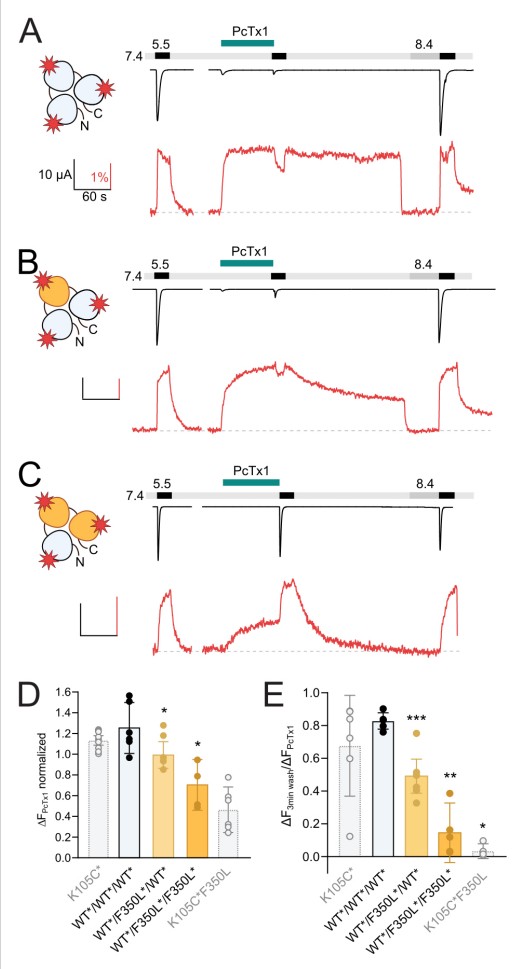

**Figure 5.** One F350L subunit is sufficient to destabilize the psalmotoxin-1 (PcTx1)-induced extracellular domain (ECD) conformation. (**A**) Representative voltage-clamp fluorometry (VCF) trace of 300 nM PcTx1 application to a concatemeric construct labeled at K105C* in all three subunits (red star) and subsequent washout for 3 min with pH 7.4 and 40 s pH 8.4. (**B**) Same as in (**A**) but one subunit carries a F350L mutation. (**C**) Same as in (**A**) but with two subunits carry a F350L mutation. (**D**) Comparison of the PcTx1-induced change in the fluorescence signal between the different concatemeric constructs shown in (**A**–**C**). Results from non-concatenated channels are indicated for comparison (shown in light gray). (**E**) Comparison of the fluorescence intensity after a 3 min washout relative to the intensity upon PcTx1 application. Results from non-concatenated channels are indicated for comparison (shown in light gray). All scale bars represent 10 μA (black vertical), 60 s (black horizontal), 1% (red vertical). In (**D**) and (**E**), error bars represents 95CI, unpaired Mann–Whitney test to neighboring bar on the left, *p<0.05, **p<0.005, ***p<0.0005.

The online version of this article includes the following source data and figure supplement(s) for figure 5:

**Source data 1.** VCF data from concatemeric mASIC1a

*Figure 5 continued on next page*

*Figure 5 continued*

of PcTx1 application and washout as shown in *Figure 5*.

**Figure supplement 1.** Voltage-clamp fluorometry (VCF) data of pH-dependent desensitization and fluorescence response of concatemers.

**Figure supplement 1—source data 1.** VCF data from concatemeric mASIC1a of the pH–dependent changes in fluorescence and SSD, as seen in *Figure 5—figure supplement 1*.

used for labeling (e.g., *Figure 1E and G*). This leaves the question if the conformational rearrangements reflect a transition into a different functional state or are related to more local PcTx1-induced conformational changes. The phenotype of the PcTx1-induced fluorescence change resembles the one observed for SSD (*Figure 1E*, *Figure 1—figure supplement 1A*), and we detect good agreement between the $pH_{50\_fluorescence}$ with the $pH_{50\_SSD}$ in all variants tested here (*Figure 1—figure supplement 1B*, *Figure 5—figure supplement 1*, *Supplementary file 1b*). This could suggest that the fluorescence change upon PcTx1 application reflects the transition of the channels from a closed to an SSD state upon PcTx1 application. However, this hypothesis is contradicted by several observations. First, during the washout of PcTx1 in K105C* or V80C*, the sustained fluorescence signal indicates a long-lived conformation not observed after SSD, and the channels can already be fully activated even before the fluorescence signal fully returns to baseline (*Figure 1E and G*, *Figure 1—figure supplement 2A*, *Figure 2A*, *Figure 2—figure supplement 1A*). Secondly, in the K105C*F350L and V80C*F350L variants, PcTx1 still introduces comparable conformational changes as seen for K105C* or V80C*, respectively, yet no functional inhibition could be observed (*Figure 3G and H*), making it unlikely that channels have ever entered SSD. We therefore favor the notion that labeling allows us to track conformational changes that do not simply report on a known functional transition, but rather reflect unique ECD conformational alterations induced by PcTx1.

## Origin of PcTx1-induced conformational changes

VCF experiments on K105C* and V80C* showed slow-onset PcTx1-induced fluorescence changes, indicating changes in the local environment of the fluorophore attached to the finger or palm domain, respectively. As outlined above, we deem it unlikely that the fluorescence changes are brought on by a direct interaction of PcTx1 with the fluorophore as the fluorescence signal follows opposite directions depending on the position

## Identification of three distinct binding modes for PcTx1

Our data suggest the presence of three distinct binding modes for PcTx1: a 'Loose,' a tight 'Global,' and a tight 'ECD$_{only}$.' We propose PcTx1 to bind loosely to the ASIC1a high-pH closed state that remains in the resting conformation even in the presence of PcTx1 (*Figure 2B and D*). This association does not result in an ECD conformational change, possibly due to the acidic pocket remaining in an expanded configuration (*Yoder et al., 2018*). Yet its physical nature is evident from the slight but consistent downward deflection in the fluorescence signal observed with

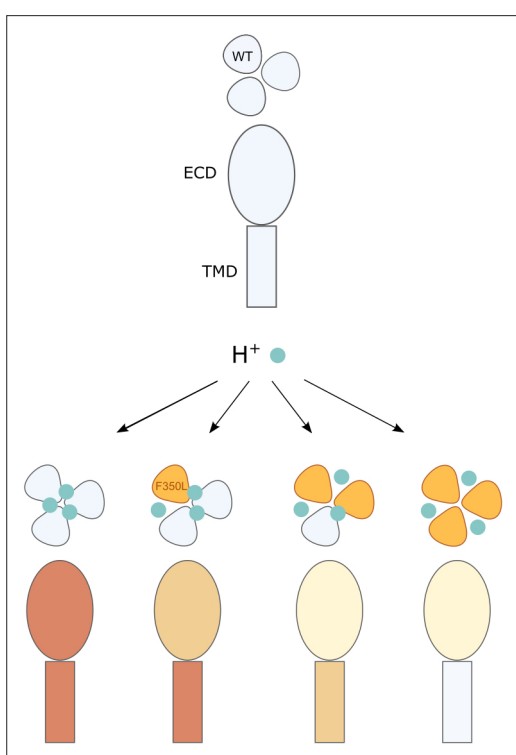

**Figure 6.** Illustration of the conformational and functional stoichiometry of psalmotoxin-1 (PcTx1) binding. Schematic representation of a side view of acid-sensing ion channel 1a (ASIC1a) extracellular domain (ECD) and transmembrane domain (TMD) and top view of the three subunits and consequences of PcTx1 (teal) binding at neutral/low pH (as in *Figure 2D*) and with the F350L mutation (orange) in 0–3 subunits. The side view coloring shows the decreasing stability of the PcTx1-induced 'ECD$_{only}$' state with increasing number of F350L-containing subunits, and the decreasing inhibitory effect on the pore. In channels with a single F350L subunit, only the PcTx1-induced conformational state of the ECD is affected, while the TMD behaves WT-like.

PcTx1 application at pH 8.0, as well as the fact that it increases the proton sensitivity of the channel (*Figure 2B*, middle panel, *Figure 2C*, *Figure 2—figure supplement 1C*). Importantly, this association of PcTx1 is readily reversible as long as the channel remains in a closed state, possibly as a result of unfavorable conformational and/or electrostatic interactions (*Figure 2B*, right panel, *Figure 2C*).

In the 'Global' mode, PcTx1 induces an alteration of the ECD conformation and modifies gating, likely through tight coupling to the channel pore. Since PcTx1 acts as an ASIC1a gating modifier, its functional consequences are pH dependent and can be inhibitory or result in enhanced activation. When PcTx1 is applied at neutral pH, the channel enters a 'Global' mode that leads to subsequent inhibition of pH 5.5 currents. By contrast, enhanced activation becomes more apparent when pH 7.0 is used as a subsequent stimulus instead: while pH 7.0 does not activate the channel directly, prior exposure to PcTx1 reveals the enhanced activation that decreases during washout (*Figure 2—figure supplement 1C*). The contribution of the left shift in activation is most obvious when the pH is subsequently changed to pH 8.4 before pH 7.0 exposure, which then reveals a return to the 'Global' state with enhanced activation.

Interestingly, an earlier study reported prolonged functional effects of PcTx1 even after washout (*Escoubas et al., 2000*), possibly due to slow dissociation (*Dawson et al., 2012*), indicating that the 'Global' binding mode can be long-lived. Here, we further observe conformational ECD changes that last substantially beyond the time frame of functional inhibition (*Figure 2A*). To our knowledge, this long-lasting 'ECD$_{only}$' binding mode has not been characterized before. Several observations indicate that PcTx1 remains bound in a distinct binding mode, such as the persistent conformational 'memory' of the 'ECD$_{only}$' state that appears to remain even after several functional and conformational transitions (e.g., *Figure 1E*, *Figure 1—figure supplement 2A*), the enhanced activation of the pH 7.0 current long after PcTx1 exposure (*Figure 2—figure supplement 1C*), and the slow dissociation constant described previously (*Dawson et al., 2012*). However, we cannot ultimately rule out that PcTx1 unbinds and leaves the channel in a distinct resting state, or that membrane-associated PcTx1 diffuses to the binding site (*Gupta et al., 2015*). Additionally, a heterogenous distribution of modes is likely co-existing across the population of receptors expressed on the cell surface, thereby further complicating measurements and interpretations. Lastly, we would like to point out that future studies may be capable of uncovering additional binding modes, for example, those associated with PcTx1-induced potentiation (*Cristofori-Armstrong et al., 2019*).

The potential physiological implications of this long-lasting conformational state are evident from our observation that it affects the activity of the endogenous neuropeptide BigDyn. And since increased BigDyn levels have been linked to acidosis-induced neuronal cell death (*Sherwood and Askwith, 2009*), this long-lasting interference of the 'ECD$_{only}$' conformation on the ASIC–BigDyn interaction could contribute to neuroprotective effects and provide a conceptually new route for ASIC-targeting therapeutics. Interestingly, such functionally silent states and conformations have previously been suggested to occur also in other ligand-gated ion channels (*Dahan et al., 2004*; *Pless et al., 2007*; *Pless and Lynch, 2009*; *Munro et al., 2019*) and could, at least in theory, be used to modulate protein–ligand and protein–protein interactions. We further speculate that long-lasting conformational changes could also interfere with recently proposed metabotropic functions of ASIC1a by altering intracellular protein–protein interactions (*Wang et al., 2015*; *Wang et al., 2020*), although this will require additional verification.

## The F350L mutation destabilizes PcTx1-induced conformational states

Consistent with previous work (*Sherwood et al., 2009*; *Saez et al., 2015*), we observed that although the F350L mutation shifts the PcTx1 IC$_{50}$ by about three orders of magnitude, ASIC1a is not rendered completely insensitive towards PcTx1 modulation. Reduced PcTx1 inhibition could, in part, result from the reduced pH sensitivity of the F350L variant, in line with the critical role of pH on PcTx1 modulation observed for human and rat ASIC1a (*Cristofori-Armstrong et al., 2019*). A mutant whose pH$_{50}$ is shifted towards higher proton concentrations (right-shifted) would show less inhibition even if PcTx1 sensitivity was maintained. However, our VCF experiments rule out decreased pH sensitivity as the main cause for reduced PcTx1 sensitivity since we corrected for the shifts in SSD and pH 5.5 is likely saturating for activation (*Borg et al., 2020*). Applied at pH 7.4, 300 nM PcTx1 (below the IC$_{50}$ of the unlabeled F350L variant) does not lead to conformational changes, but does so at pH 7.3, although no functional effects are observed. These experiments indicate that PcTx1 interacts with

F350L* and introduces conformational changes despite the lack of functional consequences at these concentrations. This is in line with earlier results that suggested PcTx1 to still bind to F352L hASIC1a as pre-incubation with PcTx1 prevented the modulatory effects of BigDyn (*Sherwood and Askwith, 2009*) and PcTx1 could still be crosslinked with the mutant channel (*Braun et al., 2021*). Similarly, experiments from ASIC1a/2a concatemers of different subunit stoichiometries suggested that PcTx1 also binds to ASIC1a-2a interfaces (*Joeres et al., 2016*) despite the fact that ASIC2a subunits contain a leucine at the position corresponding to 350 in mASIC1a. Hence, the F350L mutation diminishes functional modulation by PcTx1, but not necessarily binding. Indeed, our fluorescence data indicate that the F350L mutation appears to destabilize the PcTx1-induced 'ECD$_{only}$' state, leading to faster return of the fluorescence signal to baseline levels, suggestive of a closed *apo* state.

## Stoichiometry of PcTx1 modulation

Structural data have recently provided a wealth of insight into the molecular details of where and how drugs or toxins bind to ion channels. Yet these data cannot directly inform on the ligand–receptor stoichiometry required for the full functional effect. For example, cryo-EM structures of homotetrameric KCNQ channels suggest a stoichiometric binding of the antiepileptic small-molecule drug retigabine (i.e., one drug per each of the four subunits), while functional work shows that binding of a single retigabine molecule is sufficient to elicit the full functional effect of the channel activator (*Yau et al., 2018*; *Li et al., 2021a*; *Li et al., 2021b*). Similarly, binding of a single alpha-bungarotoxin molecule to the heteromeric nAChR pentamer is sufficient for channel inhibition despite two molecules being present in a cryo-EM co-structure (*daCosta et al., 2015*; *Rahman et al., 2020*).

Here, we show that concatemers with homomeric subunit composition display comparable pH sensitivities for activation and SSD, along with similar sensitivity to PcTx1 compared to their homotrimeric counterparts. Functionally, we demonstrate that two WT ASIC1a subunits are sufficient for WT-like PcTx1 sensitivity at physiological pH. Our VCF results, however, revealed that a single F350L subunit is sufficient to destabilize the PcTx1-induced ECD conformation, which then further decreases in channels containing two or three mutant subunits.

We would like to emphasize that we cannot categorically rule out nonlinear contributions of individual PcTx1 molecules to the net fluorescent signal. Yet we deem this possibility unlikely since the stability of the 'ECD$_{only}$' mode decreases in a roughly linear manner with increasing numbers of mutant subunits.

Together, the proposed binding modes emphasize that the conformational changes of the ECD are partially decoupled from the TMD: interfering with a single PcTx1 binding site is sufficient to alter the stability of the ECD conformation, while the pore (or the coupling to it) translates to closed channels only if two or more PcTx1 binding sites are intact (*Figure 6*).

## Conclusions

The conformational landscape of ion channels is likely to far exceed the conformations uncovered by structural or functional studies as it is often challenging to capture such states in compelling ways. By combining concatemeric construct design with VCF, we provide insight into functionally silent conformational changes of the mASIC1a ECD during PcTx1 modulation and assess the contribution of individual subunits to the overall conformational state and on the functional state of the pore. We found that PcTx1 can bind to mASIC1a with at least three distinct binding modes. In addition, PcTx1 induces a long-lived conformational state in the ASIC1a ECD that is uncoupled from the pore but alters ASIC1a pharmacology. Lastly, the work presented here, to our knowledge, constitutes the first use of concatemeric multisubunit protein assemblies engineered for VCF recordings, thus offering the possibility to label a defined subset of subunits to unveil potential asymmetric gating and binding events, such as those proposed for a variety of ligand-gated ion channels (*Jasti et al., 2007*; *Baconguis and Gouaux, 2012*; *Mowrey et al., 2013*; *Zhu et al., 2018*; *Guros et al., 2020*; *Zhang et al., 2021*; *Bergh et al., 2021*).

# Materials and methods

## Molecular biology

The complementary DNA (cDNA) encoding mouse ASIC1a (mASIC1a) was used as previously described (*Lynagh et al., 2017*). Site-directed mutagenesis was performed using PfuUltraII Fusion polymerase (Agilent) and custom DNA mutagenesis primers (Eurofins Genomics). All sequences were confirmed by sequencing of the full coding frame (Eurofins Genomics or Macrogen). cDNAs were linearized with EcoRI, and capped cRNA was transcribed with the Ambion mMESSAGE mMACHINE SP6 kit (Thermo Fisher Scientific).

## ASIC1a concatemer design

Mouse ASIC1a concatemeric constructs were generated as described previously (*Lynagh et al., 2017*). Briefly, the concatemeric mASIC1a constructs were generated by cloning out the mASIC1a insert from the pSP64 vector using forward and reverse primers containing additional HindIII and SalI, SalI and BamHI, or BamHI and SacI sequences, respectively (*Figure 4—figure supplement 1*). This generated three distinct inserts, *a*, *b*, and *c*, which were gel-purified using the Gel/PCR DNA fragment kit (Geneaid). The inserts were digested with their respective restriction enzymes – insert *a*: HindIII and SalI; insert *b*: SalI and BamHI; and insert *c*: BamHI and SacI. Insert *a* was ligated into the empty pSP64 vector (Promega) double-digested with HindIII and SalI. The resulting vector was then double-digested with SalI and BamHI to insert segment *b* in similar way. The *a-b* plasmid was then double-digested using BamHI and SacI and segment *c* was ligated into it. This yielded the final '*a-b-c*' mASIC1a concatemer. From this construct, WT inserts were replaced by mASIC1a inserts containing amino acid substitutions generated using the same primers and restriction sites. Full concatemer sequences were confirmed using primers recognizing the unique restriction sites.

## Electrophysiological recordings

Oocytes were surgically removed from adult female *X. laevis* and prepared as previously described (*Lynagh et al., 2017*). Oocytes were injected with 0.1–10 ng cRNA of mASIC1a (volumes between 9 and 50 nL) and incubated for 1–4 days at 18°C in Leibovitz's L-15 medium (Gibco), supplemented with 3 mM L-glutamine, 2.5 mg/ml gentamycin, and 15 mM HEPES adjusted to pH 7.6. Typically, larger amounts of mutant mASIC1a cRNA were injected compared to WT mASIC1a. For electrophysiological recordings, oocytes were transferred to a recording chamber (*Dahan et al., 2004*), continuously perfused (2.5 mL/min) with buffer containing (in mM) 96 NaCl, 2 KCl, 1.8 BaCl$_2$, 2 MgCl$_2$, and 5 HEPES, pH adjusted by NaOH or HCl. Solutions exchange was achieved using a gravity-driven 8-line automated perfusion system operated by a ValveBank module (AutoMate Scientific). In experiments where PcTx1 was used, all solutions were supplemented with 0.05% bovine serum albumin (≥98% essentially fatty acid-free, Sigma-Aldrich). Currents were recorded using microelectrodes (borosilicate capillaries 1.2 mm OD, 0.94 mm ID, Harvard Apparatus), backfilled with 3 M KCl (0.3–1.5 MΩ) and an OC-725C amplifier (Warner Instruments). The current signal was acquired at 500 Hz, filtered by a 50–60 Hz noise eliminator (Hum Bug, Quest Scientific) and digitized using an Axon Digidata 1550 Data Acquisition System and the pClamp (10.5.1.0) software (Molecular Devices). The current signal was further digitally filtered at 2.8 Hz using an 8-pole Bessel low-pass filter prior to analysis. Displayed current traces have been subjected to additional 50× data reduction.

Synthetic PcTx1 was obtained from Alomone Labs (>95% purity). BigDyn was synthesized using automated solid-phase peptide synthesis, followed by purification via reversed-phase high-performance liquid chromatography as described previously (*Borg et al., 2020*). Peptide stock solutions were prepared in MilliQ (18.2 MΩ resistivity) and stored at –20°C. Prior to recording, the peptide was diluted to the desired concentration in recording solution.

Concentration–response relationships of mASIC1a activation were determined by 20 s applications of solutions with decreasing pH values. Between each 20 s application, the oocytes were left to recover for at least 1 min in pH 7.4 solution (unless stated otherwise). The currents elicited by the 20 s application of acidic pH were normalized to the largest current size for each of individual oocyte tested. SSD concentration–response relationships were determined by exposing ASIC1a-expressing oocytes to a 20 s application of pH 5.6 (unless stated otherwise), while decreasing the resting pH in between applications of pH 5.6. In order to ensure that the observed current desensitization was not due to general current rundown, a final application of pH 5.6 was performed after a 2 min resting

period at a resting pH that resulted in saturating current responses. Traces were used for further analysis only if the final pH 5.6 application resulted in currents that were ≥80% of the same resting pH prior to the SSD protocol. For testing the effect of PcTx1 on the ASIC1a current response, channels were activated by 20 s application of pH 5.6. Hereafter, the pH solution returned to resting pH to allow channel recovery. PcTx1 was then applied at the desired pH for 1 min prior to a 20 s application of pH 5.6. We tested addition of up to 3 µM PcTx1 to pH 7.4 buffer and 300 nM PcTx1 to pH 7.3 buffer, but did not measure an appreciable change in pH. For all protocols only traces with current sizes <30 µA were used for further analysis.

## Voltage-clamp fluorometry

For VCF, 5–20 ng of mASIC1a mRNA or 50 ng for concatemeric constructs were injected into oocytes that were then incubated for 2–7 days. On the day of the recording, oocytes w ere labeled by incubating them for 30 min with 10 µM Alexa Fluor 488 C5-maleimide (Thermo Fisher Scientific) at room temperature in OR2 solution containing (in mM) 82.5 NaCl, 2 KCl, 1 $MgCl_2$, pH 7.4, subsequently washed twice with OR2, and stored in the dark until further use. Concatemeric constructs were kept in the dark for several hours as it tended to reduce background fluorescence. VCF recordings were done as previously described (*Borg et al., 2020*) using an inverse microscope (IX73 with LUMPlanFL N ×40, Olympus), an Olympus TH4-200 halogen lamp as light source, and using a standard GFP filter set (Olympus). The emission was detected using a P25PC-19 photomultiplier tube (Sens-Tech) and photon-to-voltage converter (IonOptix). Current and fluorescence signals were acquired, filtered, digitized, and digitally filtered as described for the other electrophysiology recordings.

For SSD and pH-dependent fluorescence measurements, running buffer at pH 7.6 was used. Labeled channels were activated for 20 s using pH 6.0, washed for 60 s with pH 7.6 buffer, exposed for 60 s to buffer at conditioning pH, and then directly activated again (*Figure 1—figure supplement 1A*). Control activations at pH 6.0 were conducted between each SSD measurement. For PcTx1 responses, running buffer at 0.3 pH units above the established $pH_{50\_fluorescence}$ was used, unless stated otherwise. First channels were activated for 20 s with pH 5.5, washed for 60 s with running buffer before applying 300 nM PcTx1 for 80 s. Washout protocols either included 3 min exposure to running buffer, pH 8.4 buffer, or a mix of the two (20 s running buffer, 60 s pH 8.4, 40 s running buffer, 60 s pH 8.4). In experiments where PcTx1 was applied at pH 8.0 for 80 s, channels were subsequently either directly exposed to pH 7.4 or activated first with pH 5.5 for 20 s, or pH 8.0 for 80 s before switching to pH 7.4. Fluorescence levels 60 s after switching to pH 7.4 were compared to the levels reached during activation with pH 5.5. For BigDyn experiments, 1 µM BigDyn was applied at pH 7.4 either for 60 s before PcTx1 application or for 120 s once 300 nM PcTx1 had been washed off with pH 7.4 for 120 s.

## Surface protein purification

Oocyte surface proteins were purified using the Pierce Cell Surface Protein Isolation Kit (Thermo Fisher Scientific), modified here for oocytes as previously described (*Lynagh et al., 2017*). In brief, 30 oocytes were injected with RNA of a single mASIC1a construct. After 2 days of incubation, the oocytes were washed with phosphate buffered saline (PBS) and incubated for 25 min at room temperature with sulfo-NHS-SS-biotin. Hereafter, the reaction was quenched and oocytes were washed with PBS. The oocytes were lysed, centrifuged, and the clarified supernatant was transferred to a spin column containing equilibrated NeutrAvidin Agarose and incubated for 1 hr at room temperature under gentle agitation. The flow-through was discarded, and the NeutrAvidin Agarose was washed and centrifuged. The wash procedure was repeated for a total of three times. Prior to elution, 200 µL SDS-PAGE sample buffer (62.5 mM Tris-HCl, 1% SDS, 10% glycerol, pH 6.8) containing 50 mM dithiothreitol (DTT) was added and the column was incubated for 60 min at room temperature under agitation. The protein was then eluted by centrifugation. The concentration of protein was determined using the NanoDrop One$^c$ UV-Vis spectrophotometer, and the samples were stored at –20°C.

## Western blot

Oocyte surface protein samples were denatured for 5 min at 90°C in LDS sample buffer containing reducing agent (50 mM DTT) (NuPAGE, Thermo Fisher Scientific). Protein samples (10–120 µg, dependent on the construct in order to obtain approximately equal band intensity), Novex Sharp pre-stained standard (Thermo Fisher Scientific), were separated by SDS-PAGE using a 4–12% Bis-Tris protein gel

(NuPAGE, Thermo Fisher Scientific). Protein bands were transferred to a polyvinylidene difluoride (PVDF) membrane (0.45 µm, Thermo Fisher Scientific) in transfer buffer (NuPAGE, Thermo Fisher Scientific) containing 10% methanol (MeOH). After transfer, the PVDF membrane was rinsed with Tris buffered saline 20 mM Tris, 150 mM NaCl, pH 7.6 with 0.1% Tween-20 added (TBST), and subsequently blocked with 0.5% non-fat dried milk (AppliChem) in TBST using the Snap i.d. 2.0 Protein Detection System (Millipore). The blot was then incubated for 10 min at room temperature with a 1:1000 dilution of rabbit polyclonal anti-ASIC1a antibody (OSR00097W, Thermo Fisher Scientific) in blocking buffer followed by 10 min incubation at room temperature with goat anti-rabbit IgG/horseradish peroxidase-conjugate (1:10,000 dilution of 1 mg/mL stock in blocking buffer; A16110, Thermo Fisher Scientific). The blot was developed using enhanced chemiluminescence detection reagents (Pierce ECL Western Blotting Substrate, Thermo Fisher Scientific) and visualized using a PXi Touch Gel Imaging System (Syngene).

## Data analysis

Data analyses were performed in Prism (8.0; GraphPad Software). Fluorescence baseline was adjusted for all fluorescence traces before analysis. For pH–response data, peak amplitudes were normalized to control activations, plotted as a function of pH, and fitted with the Hill equation constrained at min = 0 (and max = 1 for TEVC data) for each recording. The resulting $EC_{50}$ and Hill values were averaged to give the reported means ± 95 CI in the tables. For PcTx1 inhibition experiments, current responses after PcTx1 treatment were normalized to the current responses without PcTx1 incubation for each oocyte. The average data for the different PcTx1 concentrations were fitted using a Hill equation constrained at min = 0 and max = 1. For display in figures, a single fit to the average normalized responses (±95 CI) is shown. The change in fluorescence upon PcTx1 application was normalized to the neighboring pH 5.5 application. Washout of PcTx1 in VCF experiments was reported as the level of fluorescence under running buffer at the end of the 3 min washout relative to the PcTx1-induced fluorescence change. Fluorescence signal intensity was reported as percentage change relative to the baseline fluorescence level.

All bar diagrams and summarized data points are presented as mean ± 95 CI unless stated otherwise, and the number of replicates (n) represents individual experimental oocytes. Results were obtained from at least two batches of oocytes. An unpaired Mann–Whitney test was used to compare two groups. Multiple comparisons were made with one-way analysis of variance with Dunnett's comparison to a control value (i.e., washout using running buffer). A significance level of $p < 0.05$ was applied for all analyses. All graphs and illustrations were made in Prism (8.0, GraphPad Software) and Illustrator CC (Adobe), and structure rendering was done using ChimeraX (*Pettersen et al., 2021*).

## Acknowledgements

We acknowledge funding from the Lundbeck Foundation (R303-2018-3030 to SAH and R313-2019-571 to SAP), the Brødrene Hartmanns Fond, and the European Union's Horizon 2020 research and innovation program under the Marie Skłodowska-Curie grant agreement no. 834274 (to SAH). We thank Drs. Han Chow Chua and Samuel G Usher for comments on the manuscript.

## Additional information

### Competing interests

Stephan A Pless: Reviewing editor, *eLife*. The other authors declare that no competing interests exist.

### Funding

| Funder | Grant reference number | Author |
| --- | --- | --- |
| Lundbeckfonden | R303-2018-3030 | Stephanie A Heusser |
| H2020 Marie Skłodowska-Curie Actions | 834274 | Stephanie A Heusser |
| Hartmann Fonden | | Stephanie A Heusser |

| Funder | Grant reference number | Author |
|--------|------------------------|--------|
| Lundbeckfonden | R313-2019-571 | Stephan A Pless |

The funders had no role in study design, data collection and interpretation, or the decision to submit the work for publication.

## Author contributions

Stephanie A Heusser, Conceptualization, Data curation, Formal analysis, Funding acquisition, Investigation, Methodology, Validation, Visualization, Writing – original draft, Writing – review and editing; Christian B Borg, Conceptualization, Data curation, Formal analysis, Investigation, Methodology, Validation, Visualization, Writing – original draft, Writing – review and editing; Janne M Colding, Methodology, Resources, Validation; Stephan A Pless, Conceptualization, Funding acquisition, Investigation, Methodology, Project administration, Supervision, Writing – original draft, Writing – review and editing

## Author ORCIDs

Stephanie A Heusser ⬤ http://orcid.org/0000-0003-3224-4547
Christian B Borg ⬤ http://orcid.org/0000-0002-8974-2670
Stephan A Pless ⬤ http://orcid.org/0000-0001-6654-114X

## Decision letter and Author response

Decision letter https://doi.org/10.7554/eLife.73384.sa1
Author response https://doi.org/10.7554/eLife.73384.sa2

# Additional files

## Supplementary files

• Supplementary file 1. Data tables. (a). Effect of psalmotoxin-1 (PcTx1) on activation and steady-state desensitization (SSD) of WT and F350L mASIC1a. Data are reported as mean with 95 CI in brackets. (b). SSD and pH-dependent changes in fluorescence for labeled mASIC1a constructs. Data are reported as mean with 95 CI in brackets. (c). Resulting fluorescence change upon application of 300 nM PcTx1 for mASIC1a-labeled constructs. Data are reported as mean with 95 CI in brackets. (d). Change in fluorescence after a 3 min washout of 300 nM PcTx1. Data are reported as mean with 95 CI in brackets. (e). Comparison of current and fluorescence in experiments where 300 nM PcTx1 were applied at pH 7.4. The channels then underwent three 1 min washouts each followed by pH 5.5 stimulus or a single 3 min washout with pH 7.4 followed by a pH 5.5 stimulus. The final pH 5.5 current was normalized to the one at the beginning of the recording, and the fluorescence was analyzed at pH 7.4 right before the final pH 5.5 activation and normalized to deflection induced by PcTx1. Data are reported as mean with 95 CI in brackets. (f) Fluorescence after PcTx1 applications at pH 8.0 followed by different pH regimes (pH 5.5, 7.4, or 8.0) and a washout with pH 7.4. Fluorescence is reported 1 min into the final pH 7.4 application. Data are reported as mean with 95 CI in brackets. (g) Resulting fluorescence change upon peptide application K105C* mASIC1a normalized to the fluorescence response elicited by pH 5.5 application. Big dynorphin (BigDyn) was applied at a concentration of 1 µM, and PcTx1 was applied at a concentration of 300 nM. The fluorescence change was monitored 1 min after application for BigDyn and 30–60 s after application for PcTx1. Data are reported as mean with 95 CI in brackets. (h) Summary of PcTx1 inhibition of mASIC1a constructs. Data are reported as mean with 95 CI in brackets. (i) pH sensitivity of activation and SSD for mASIC1a constructs. Data are reported as mean with 95 CI in brackets.

• Transparent reporting form

## Data availability

Figure 1—source data 1, Figure 1—source data 2, Figure 1—figure supplement 1—source data 1, Figure 1—figure supplement 1—source data 2, Figure 2—source data 1, Figure 2—source data 2, Figure 2—source data 3, Figure 3—source data 1, Figure 3 - Source Data 2, Figure 3—source data 3, Figure 4—source data 1, Figure 4—source data 2, Figure 4—source data 3, Figure 5—source data 1 and Figure 5—figure supplement 1—source data 1 contain the numerical data used to generate the figures.

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
