## [Editor Report]

This study provides direct evidence that PcTx1, a modulator commonly used to study acid-sensing ion channels, induces a conformational change that persists long after an effect on the channel activity has dissipated. The data support this central claim of the paper and invite future investigation of the precise mechanism. The work is of general interest to those studying ion channel biophysics and pharmacology and is an exemplar of the power of combined functional and fluorescence measurements.

---

## [Decision Letter]

**Decision letter after peer review:**

Thank you for submitting your article "Conformational decoupling in acid-sensing ion channels uncovers mechanism and stoichiometry of PcTx1-mediated inhibition" for consideration by *eLife*. Your article has been reviewed by 3 peer reviewers, including Jon T Sack as Reviewing Editor and Reviewer #1, and the evaluation has been overseen by Kenton Swartz as the Senior Editor. The following individual involved in review of your submission has agreed to reveal their identity: David M Maclean (Reviewer #2).

Essential revisions:

1) Address whether PcTx1 effects on F350L could be due to inadvertent acidification of PcTx1 solutions.

2) Elaborate on when PcTx1 is bound in relation to ECD and pore conformational changes.

3) Elaborate on basis for claims that uncoupling between the ECD and pore is occurring.

A straightforward way to address these requests would be inclusion of additional results. Further details of these concerns are in the reviews, along with questions about the mechanistic implications and suggestions that we hope you will find helpful.

*Reviewer #1 (Recommendations for the authors):*

This is a beautiful study. I have one major concern: that some the effects of solutions with higher concentrations of PcTx1 on F350L may result from a more acidic pH. There is no indication that pH was checked after addition of PcTx1. The PcTx1 solutions are made from synthetic peptide, which are commonly lyophilized from acidic solution. The amount of acid in lyophilizates can vary between preparations. Even a small amount of acid in the extracellular buffer could lead to steady state desensitization. 3 pieces of the data seem consistent with this acidic concern:

1) the Hill slopes >1 for PcTx1 concentration-effect on F350L: this is consistent with H^+^ desensitization, but not PcTx1 which shows no indication of cooperativity at lower concentrations.

2) the variability of inhibition at [PcTx1] {greater than or equal to} 100 nM (Figures 3E, 4G)

3) inexplicably dramatic difference in effect with pH 7.4 vs pH 7.3 solution (Figures 3F,G)

Could result from changes in pH after addition of PcTx1 from different lots?

Due to this concern, and the potential impacts on conclusions, I would suggest assessing whether the effects of PcTx1 solutions on F350L could result from merely a more acidic pH.

If this concern is without basis, I apologize for the trouble.

Small suggestions:

Page 5 "…conformational change of the ECD upon PcTx1 binding."

Suggest changing to "…after PcTx1 binding."

Page 17 "VCF experiments rule out decreased pH sensitivity as the main cause for reduced PcTx1 sensitivity since we corrected from shifts in the SSD."

Might this statement be qualified due to unaccounted greater shifts of pH activation?

*Reviewer #2 (Recommendations for the authors):*

The PcTx1 stoichiometry (Figures 4 and 5) experiments are a valuable addition to the field. However, the long-lived effects of PcTx1 are the most intriguing aspect of the paper. It's well worth adding experiments to distinguish this novel conformation with states induced purely by pH, to more deeply contrast the proposed Global and ECDonly states and to test and refine their model.

The PcTx1-induced δ F signals observed with K105C* and V80C* are also seen with pH 6 stimulus (Bonifacio et al., 2014 Figure 1). And relatively weak pH stimulus (~ pH 6.9, Bonifacio et al., 2014 Figure 6) also produces δ F signals at these positions. PcTx1 works by increasing apparent affinity for protons (this study, Figure 1D). Very roughly, when the channel is incubated with PcTx1 at pH 7.4 it behaves as it would when alone at pH 7.1 or 7. I suspect that at either site, an SSD-style experiment from 7.4 to 7 would produce a δ F of similar direction but smaller magnitude as PcTx1 applied at 7.4 but the fluorescence would return to baseline much faster when moving back to 7.4.In other words, it would be good to compare the PcTx1+pH7.4-induced signal with that induced by a pH stimulus of similar functional consequence. This would further highlight that the conformation induced by PcTx1 at 7.4 is distinct from that induced by a functionally equivalent SSD causing stimulus.

In addition, the dissociation of Global from ECDo states could be done more rigorously through a few experiments. In the absence of PcTx1, pH 5.5 is saturating while pH 7 produces basically no current response. However, in the presence of PcTx1 pH 7 gives about 80% max response (Figure 1D). If the authors are correct that the ECDo state has uncoupled the ECD from the pore, than the pH 7 response should return back to near 0 in the ECDo state. Thus the relative response evoked by pH 7 could be used to distinguish the Global and ECDo states using the experiments in Figure 1E/G (left). After the PcTx1 application, stimulate with 7 and then repeat the stimulus 1-3 minutes into the washes. If the ECDo state is populated, the pH 7 response should be much smaller. However, if PcTx1 is still there in the Global state, the pH 7 responses should be relatively large. Alternative sweeps using pH 5.5 might be needed for normalizing purposes. The current amplitudes might be small with pH 7 stimulus but using a baseline of 7.5 or 7.6 might make the experiment more feasible. I think this would make for a much clearer distinction than the BigDyn experiment and much more compelling evidence for altered coupling (Figure 2 E,F).

The model in Figure 2D is very helpful in getting across the authors message but does imply PcTx1 is still bound in the ECDo state. I think this has problematic implications (see below). However, it can rather easily be tested by simply adding PcTx1 a second time. Using the protocol in Figure 1E/G (left), after PcTx1 incubation, pH 5.5 is applied and the current is much inhibited. pH 7.4 is applied and the fluorescence remains very different from baseline as the channels, presumably, enter the ECDo state. The model (Figure 2D) indicates PcTx1 is still bound in this state (but possibly to some distinct sub-site or altered pose?). If true, then applying PcTx1 again should do nothing. All PcTx1 sites are occupied by toxins holding the channel in the ECDo state. No new toxins can find spots so the fluorescence should be unchanged and the channels not modulated. However, if something else is going on (if PcTx1 has unbound but left the channel in a slightly different resting state for example), then a second PcTx1 application is likely to produce inhibition just as the first did. As above, I think this makes for a much more compelling distinction between Global and ECDo than the BigDyn experiment and a more explicit test of the model.

Why don't ASICs bound by PcTx1 always go into the ECDo state? If ASIC-PcTx1 complexes enter a novel long-lived non-modulated state where the toxin remains bound even well after removal from the bathing solution, then these complexes should also enter this same state when PcTX is still in the bathing solution. Fully toxin-bound channels should behave the same regardless of how much unbound toxin is floating around. From Results 4th para, all that is required for ECDo occupancy is PcTx1 plus "conditions that temporarily activate and/or desensitize the channel". Thus unless I am much mistaken, the proposed ECDo state should form during typical SSD or pH response curves in the continual presence of PcTx1. You should get ECDo states in the pH response curve in Figure 1C (lower) for example. The pH 7.6 incubation followed by pH 5.5 challenge should trigger the formation of the ECDo state which should preclude modulation by PcTx1 at pH 7.4, and the current should come back. But instead you get inhibition. I think a stronger interpretation of this data is that PcTx1 unbinding enables a distinct resting state (as seen with δ F) with dampened fluorescence responses to BigDyn. The strength of ECD-pore coupling in this state, and it's occupancy by PcTx1, needs further experimental examination of the type suggested above.

A lot of the data might be explained by PcTx1 fluorescence and modulation have different stoichiometries. It could be that one bound PcTx1 induces a pretty big δ F (or even a global change in the trimer) but doesn't really provide much modulation. A second or third PcTx1 might produce smaller (or maybe even no) change in deltaF but the second and third are needed for channel modulation. Once you have a trimer saturated with PcTx1, sooner or later one PcTx1 and then a second will dissociate, leaving you with a channel that is no longer modulated, but that last residing PcTx1 keeps the fluorescence relatively high (or low for V80). This aligns nicely with their Figure 4G data. The only objection I see to this 'One PcTx1 changes Fluorescence, two change gating' model is the VCF in Figure 5. In 5C, two binding sites are disrupted. If the above model is correct, a single PcTx1 should give a high deltaF.

Is it possible the pH's for this experiment in 5C are not optimal? They use 7.4 all the way through. But the SSD for the concatemers in 5C is shifted acidic by ~ 0.2 pH units. If they repeat 5C using 7.2 as the basal pH, I wager they would get a pretty big δ F from just that one PcTx1 binding. This would further support the 'One PcTx1 for the F, two for the gating' idea.

Ultimately, I think this is a very interesting and useful paper which will make a sizeable impact in the field and completely solving the mechanism of the PcTx1-induced δ F shouldn't be a requirement for publication.

*Reviewer #3 (Recommendations for the authors):*

1. The main claim in this study is that changes in the magnitude of fluorescent signals generated by fluorescent molecules attached to ASIC1a arise from a global conformational change of the ECD induced by binding of PcTx1 to the channel. Since the most important evidence relies on the interpretation of this measurement, I would like further explanation on the following points:

a) The two residues chosen for placing the fluorescent labels (V80 and K105) are entirely exposed to water (resting, open, desensitized) in all the structures of human and chicken ASIC available today. Therefore, it is expected that the environment of those sites would not change significantly. Can the authors explain the reason of choosing specifically those two residues?

b) What is the mechanism of the small decrease in fluorescence seen after PcTx1 and pH 5.5 in Figure 1A and Figure 2A. Activation prior to the addition of PcTx1 produces a large increase in fluorescence in channels with K105C* but produces a small negative deflection when channels are activated after addition of the toxin; i.e., is the fluorescent signal produced only by desensitization but activation is silent? Is the downward deflection an artifact or reflects a physicala processs dismissed in the study? Please, provide an explanation.

c) Figure 2A the fluorescent signal shows persistence of 'a long-lasting desensitized conformation' induced by PcTx1 but channels are functional not desensitized. Please, explain what factor(s) determines the shift from one mode the other and whether in a single cell there is heterogeneity of modes.

d) The protocol in the first trace of Figure 2A shows multiple activations with low pH. Have the authors considered whether channel activation plays a role in regaining function while the fluorescent signal remains high. In this case the protocol will differ by waiting 3 min after PcTx1 to apply the first activation.

e) The authors write in the first paragraph of results: 'During a 3 min washout out with pH 7.4 solution, the fluorescence signal displayed only a modest decrease, while exposure …' Do they mean that the toxin was washed out during that period? PcTx could remain bound to the channel for much longer than 3 min as shown by Dawson et al., (DOI: 10.1038/ncomms1917) using SPR at pH 6.3 they measured KD = 1.79x10-9 M, Kon=7.99x105 M s, and Koff=1.4x10-3 that is a tao ~12 min, suggesting that after 3 min there is almost no washout of the toxin. The presence or absence of the toxin attached to the channels needs to be verified with additional experiments. Another caveat, is that affinity of the toxin may be also be pH dependent, complicating the interpretation of changes in fluorescence attributed to pH.

2. The MN focuses in a narrow and highly specialized topic requiring a great deal of familiarity with the subject. The Introduction could make a better job in this regard. The following sentences in the Abstract are unintelligible prior to reading the whole MN: ‘The loose and ECDonly modes are decoupled from the pore, with the latter inducing a long-lived ECD conformation that reduces the crossd of an endogenous neuropeptide. This long-lived conformational state is proton-dependent and can be destabilized by the F350L mutation”. The MN will benefit from deeper description of modes, how are generated, and what underlies the long lived ‘desensitization’ while regaining response to protons.

3. It would be informative to add to Figure 1B and 1C the fluorescence traces below the currents of activation and SSD. Here, the concentration of PcTx1 is 30 nM, ten-fold lower than in subsequent experiments (300 nM) but is sufficient to shift to the right the pH50ssd and pH50a by one pH unit. The question is whether the magnitude of the fluorescence signal is smaller or remains as in the experiments done with 300 nM. Also, if the fluorescent signal persists or wanes with subsequent activations.

---

## [Author Response]

Essential revisions:1) Address whether PcTx1 effects on F350L could be due to inadvertent acidification of PcTx1 solutions.

We have measured the pH after adding PcTx1 to the buffer at concentrations up to 3 μM at pH 7.4 and 300 nM at 7.3 and we did not observe any measurable acidification (see below for details).

2) Elaborate on when PcTx1 is bound in relation to ECD and pore conformational changes.

We have now added new experimental data based on the suggestions of reviewer 3 and have made changes to the figures and text in several sections to elaborate on the different PcTx1 binding modes more clearly. We hope the additional data and explanations sufficiently address this point.

3) Elaborate on basis for claims that uncoupling between the ECD and pore is occurring.

We have made several changes to the text and figures that should clarify the uncoupling proposed by us.

A straightforward way to address these requests would be inclusion of additional results. Further details of these concerns are in the reviews, along with questions about the mechanistic implications and suggestions that we hope you will find helpful.

We are grateful for the very constructive feedback and we have now addressed all reviewer comments in detail below. We believe the additional experiments, along with the rewording of several sections have made the narrative clearer.

Reviewer #1 (Recommendations for the authors):This is a beautiful study. I have one major concern: that some the effects of solutions with higher concentrations of PcTx1 on F350L may result from a more acidic pH. There is no indication that pH was checked after addition of PcTx1. The PcTx1 solutions are made from synthetic peptide, which are commonly lyophilized from acidic solution. The amount of acid in lyophilizates can vary between preparations. Even a small amount of acid in the extracellular buffer could lead to steady state desensitization. 3 pieces of the data seem consistent with this acidic concern:1) The Hill slopes >1 for PcTx1 concentration-effect on F350L: this is consistent with H^+^ desensitization, but not PcTx1 which shows no indication of cooperativity at lower concentrations.2) The variability of inhibition at [PcTx1] {greater than or equal to} 100 nM (Figures 3E, 4G)3) Inexplicably dramatic difference in effect with pH 7.4 vs pH 7.3 solution (Figures 3F, G)Could result from changes in pH after addition of PcTx1 from different lots?Due to this concern, and the potential impacts on conclusions, I would suggest assessing whether the effects of PcTx1 solutions on F350L could result from merely a more acidic pH.If this concern is without basis, I apologize for the trouble.

This a good point and a reasonable concern. We have measured the pH before and after adding PcTx1 at concentrations up to 3 μM at pH 7.4 and 300 nM at 7.3 and saw no measurable changes in pH. This was performed with two different PcTx1 stocks from commercial sources. We therefore conclude that the abovementioned effects are not, or only marginally, due to PcTx1-induced acidification of the buffers. We have now added a statement in the method section to explicitly address this point.

Small suggestions:Page 5 “…conformational change of the ECD upon PcTx1 binding.”Suggest changing to “…after PcTx1 binding.”

We thank the reviewer for the suggestion. The sentence has now been changed in the manuscript.

Page 17 “VCF experiments rule out decreased pH sensitivity as the main cause for reduced PcTx1 sensitivity since we corrected from shifts in the SSD.”Might this statement be qualified due to unaccounted greater shifts of pH activation?

This could indeed be a concern. Based on previously published data of the K105C variant (PMID: 32165542), we know that pH 5.5 is a super-saturating proton concentration for activation. Given the relatively small shifts in SSD of the crossd variants, we would thus also expect pH 5.5 be saturating for those. We have added additional text on page 15 to directly address this point.

Reviewer #2 (Recommendations for the authors):The PcTx1 stoichiometry (Figures 4 and 5) experiments are a valuable addition to the field. However, the long-lived effects of PcTx1 are the most intriguing aspect of the paper. It’s well worth adding experiments to distinguish this novel conformation with states induced purely by pH, to more deeply contrast the proposed Global and ECDonly states and to test and refine their model.The PcTx1-induced δ F signals observed with K105C* and V80C* are also seen with pH 6 stimulus (Bonifacio et al., 2014 Figure 1). And relatively weak pH stimulus (~ pH 6.9, Bonifacio et al., 2014 Figure 6) also produces δ F signals at these positions. PcTx1 works by increasing apparent affinity for protons (this study, Figure 1D). Very roughly, when the channel is incubated with PcTx1 at pH 7.4 it behaves as it would when alone at pH 7.1 or 7. I suspect that at either site, an SSD-style experiment from 7.4 to 7 would produce a δ F of similar direction but smaller magnitude as PcTx1 applied at 7.4 but the fluorescence would return to baseline much faster when moving back to 7.4.In other words, it would be good to compare the PcTx1+pH7.4-induced signal with that induced by a pH stimulus of similar functional consequence. This would further highlight that the conformation induced by PcTx1 at 7.4 is distinct from that induced by a functionally equivalent SSD causing stimulus.

Yes, this is a correct observation. The application of pH 7.0 leads to a similar ΔF and to similar inhibition as PcTx1. As correctly pointed out by the reviewer, and in contrast to the PcTx1 effect, the pH 7.0-induced effect washes out almost immediately. The pH-dependent effect on the fluorescence is shown in Figure 1 figure supplement 1A and for additional clarification we now added an inset of the below pH 7.0 application without a subsequent pH 5.5 stimulus. We have also provided additional text on page 4 and 14 to emphasize this observation

In addition, the dissociation of Global from ECDo states could be done more rigorously through a few experiments. In the absence of PcTx1, pH 5.5 is saturating while pH 7 produces basically no current response. However, in the presence of PcTx1 pH 7 gives about 80% max response (Figure 1D). If the authors are correct that the ECDo state has uncoupled the ECD from the pore, than the pH 7 response should return back to near 0 in the ECDo state. Thus the relative response evoked by pH 7 could be used to distinguish the Global and ECDo states using the experiments in Figure 1E/G (left). After the PcTx1 application, stimulate with 7 and then repeat the stimulus 1-3 minutes into the washes. If the ECDo state is populated, the pH 7 response should be much smaller. However, if PcTx1 is still there in the Global state, the pH 7 responses should be relatively large. Alternative sweeps using pH 5.5 might be needed for normalizing purposes. The current amplitudes might be small with pH 7 stimulus but using a baseline of 7.5 or 7.6 might make the experiment more feasible. I think this would make for a much clearer distinction than the BigDyn experiment and much more compelling evidence for altered coupling (Figure 2E,F).

We thank the reviewer for these interesting suggestions. In the manuscript, we primarily focus on the inhibitory effect of PcTx1 (as indicated by the title) that are due to the left-shift of the SSD curve. By contrast, we did not focus on the change of the activation curve (and possibly related binding modes), nor the potentiating effect of PcTx1 observed by others (PMID: 30849303) in more detail.

However, since we were also curious about the outcome of the suggested experiments, we conducted additional VCF recordings on K105C*. Before discussing the results, we would like to point out that a direct comparison between the data in Figure 1D and the conditions that allow the best visualization of the ECD_only_ state is not trivial: first, the labelled construct and the WT differ in their sensitivity to PcTX1 and pH, and second, the pH conditions between the VCF recordings in e.g. Figure 1E and the ones in Figure 1B are not the same.

Application of pH 7.0 to K105C* does not lead to any currents. However, application of PcTx1 at pH 7.4 and subsequent pH 7.0 stimulus leads to measurable channel activation. We think that this current represents the net result of two distinct phenomena: on the one hand, PcTx1 *increases* the current response due to the left-shift of the activation curve, but on the other hand, it also *inhibits* the pH 7.0-induced current via the left-shift of the SSD curve. In Figure 1B, D, the activation at pH 7.0 is more pronounced, since PcTx1 was administered at pH 7.9, thus leading to a smaller contribution of SSD (in Figure 2B, middle panel, we show that application of PcTx1 at pH 8.0 to K105C* leads to subsequent activation by pH 7.4).Once the channel is in the ECD_only_ state, repeated pH 7.0 stimulation leads to decreasing currents. Surprisingly, temporary exposure to pH 8.4, followed by a subsequent pH 7.0 stimulus leads to about 50% activation. This indicates that (a) PcTx1 can still impact activation, even if the inhibitory effect is lost and (b) that the transition from the ‘Loose’ to the 'Global’ state is possible despite the absence of PcTx1 in the extracellular solution. We have added the experiment to Figure 2—figure supplement 1C and described it on pages 6 and 15. We have also further elaborated around the fact that there might be additional PcTx1 binding modes. We have highlighted that effects such as the potentiation exerted by PcTx1 add further complexity to the overall picture of the ASIC-PcTx1 interaction and will require additional exploration in the future.

The model in Figure 2D is very helpful in getting crosss the authors message but does imply PcTx1 is still bound in the ECDo state. I think this has problematic implications (see below). However, it can rather easily be tested by simply adding PcTx1 a second time. Using the protocol in Figure 1E/G (left), after PcTx1 incubation, pH 5.5 is applied and the current is much inhibited. pH 7.4 is applied and the fluorescence remains very different from baseline as the channels, presumably, enter the ECDo state. The model (Figure 2D) indicates PcTx1 is still bound in this state (but possibly to some distinct sub-site or altered pose?). If true, then applying PcTx1 again should do nothing. All PcTx1 sites are occupied by toxins holding the channel in the ECDo state. No new toxins can find spots so the fluorescence should be unchanged and the channels not modulated. However, if something else is going on (if PcTx1 has unbound but left the channel in a slightly different resting state for example), then a second PcTx1 application is likely to produce inhibition just as the first did. As above, I think this makes for a much more compelling distinction between Global and ECDo than the BigDyn experiment and a more explicit test of the model.

We thank the reviewer for this valuable suggestion. We have now conducted an experiment where we reapply PcTx1 to the ECD_only_ state and included the results in Figure 2G, H. The fluorescence and the observed inhibitory effect on the subsequent pH 5.5-induced current are comparable between the two PcTx1 applications. This shows that newly applied PcTx1 can bind to the ECD_only_ state and reintroduce inhibition. This further indicates that PcTx1-binding to the ECD_only_ and the Global state are distinct, as indicated by the schematic in 2D – but this does not necessarily exclude PcTx1 unbinding (see also next comment). We are now considering these possibilities in more detail in the Discussion section on page 15.

Why don’t ASICs bound by PcTx1 always go into the ECDo state? If ASIC-PcTx1 complexes enter a novel long-lived non-modulated state where the toxin remains bound even well after removal from the bathing solution, then these complexes should also enter this same state when PcTX is still in the bathing solution. Fully toxin-bound channels should behave the same regardless of how much unbound toxin is floating around. From Results 4^th^ para, all that is required for ECDo occupancy is PcTx1 plus “conditions that temporarily activate and/or desensitize the channel”. Thus unless I am much mistaken, the proposed ECDo state should form during typical SSD or pH response curves in the continual presence of PcTx1. You should get ECDo states in the pH response curve in Figure 1C (lower) for example. The pH 7.6 incubation followed by pH 5.5 challenge should trigger the formation of the ECDo state which should preclude modulation by PcTx1 at pH 7.4, and the current should come back. But instead you get inhibition. I think a stronger interpretation of this data is that PcTx1 unbinding enables a distinct resting state (as seen with δ F) with dampened fluorescence responses to BigDyn. The strength of ECD-pore coupling in this state, and it’s occupancy by PcTx1, needs further experimental examination of the type suggested above.

The ECD_only_ state can indeed bind newly applied PcTx1, shown now more explicitly with the suggested experiment in Figure 2G. For the global inhibitory state, PcTx1 is needed in the buffer, indicating distinct binding modes. However, it is indeed possible that there is a co-existing heterogenous population of different binding modes on the cell surface. We state this now more explicitly in the discussion on page 15. The question remains whether PcTx1 is bound to the ECD_only_ state or whether it leaves it in a distinct resting state. Three observations indicate that PcTx1 remains bound to the channel (a) The ‘memory’ of the ECD_only_ state persists even if the channel undergoes several functional and conformational transitions (e.g. Figure 1E). It thus is unlikely that it ‘locks’ the channel in a specific state; (b) The pH 7.0 stimulus experiments from comment #2 show that enhanced activation is still possible long after PcTx1 exposure, favoring a model where PcTx1 is still bound; (c) As pointed out by reviewer 3, Dawson *et al.,* 2012 (PMID: 22760635) have performed SPR experiments that indicate an extremely slow wash-off of PcTx1, again suggesting the presence of PcTx1 even after prolonged washouts. We hope that the added text in the discussion on page 15 covers this point sufficiently.

A lot of the data might be explained by PcTx1 fluorescence and modulation have different stoichiometries. It could be that one bound PcTx1 induces a pretty big δ F (or even a global change in the trimer) but doesn't really provide much modulation. A second or third PcTx1 might produce smaller (or maybe even no) change in deltaF but the second and third are needed for channel modulation. Once you have a trimer saturated with PcTx1, sooner or later one PcTx1 and then a second will dissociate, leaving you with a channel that is no longer modulated, but that last residing PcTx1 keeps the fluorescence relatively high (or low for V80). This aligns nicely with their Figure 4G data. The only objection I see to this 'One PcTx1 changes Fluorescence, two change gating' model is the VCF in Figure 5. In 5C, two binding sites are disrupted. If the above model is correct, a single PcTx1 should give a high deltaF.

Yes, we agree with the reviewer's observation that the PcTx1 fluorescence and modulation have different stoichiometries; this is what we refer as the decoupled PcTx1 effect. However, it appears that we arrive at this proposition from different angles. If we understand correctly, the reviewer brings up the possibility that binding of PcTx1 might have non-linear contributions to the magnitude of the fluorescent signals.

Conversely, we base our argumentation not primarily on the size of the PcTx1-induced ΔF, but rather focus on its washout. After all, the F350L mutation in a homomeric trimer still leads to a comparable ΔF (Figure 3G), indicating binding and introduction of the ECD_only_ state, but the washout after 3 min is very distinct compared to that of K105C*. We thus argue that increasing numbers of F350L-containing subunits affect the stability of the ECD_only_ state (Figure 3 G,H). While at this point we cannot rule out non-linear contributions of different PcTx1 molecules to the fluorescence signal, our observation about the increasing destabilization of the ‘ECD_only_’ remains valid since it is not based on the comparison of the magnitude of the PcTx1-induced ΔF.

We now bring up this point in the discussion on page 16.

Regardless of the above, the question of the contribution of just one PcTx1 molecule on the conformation to a specific subunit (especially in combination with the F350L mutations) is certainly interesting, and one that we ultimately would like to tackle in the future by fluorescently labelling only a subset of subunits. These experiments are, however, non-trivial for a number of reasons (extensive concatemer designs, complex fluorescence signals, etc) and will require a substantial amount of additional experimental work that would exceed the scope of this manuscript.

Is it possible the pH's for this experiment in 5C are not optimal? They use 7.4 all the way through. But the SSD for the concatemers in 5C is shifted acidic by ~ 0.2 pH units. If they repeat 5C using 7.2 as the basal pH, I wager they would get a pretty big δ F from just that one PcTx1 binding. This would further support the 'One PcTx1 for the F, two for the gating' idea.

While the SSD for the non-labelled constructs is indeed shifted, the labelled constructs all have a pH_50_SSD_ and pH_50_fluorescence_ of around 7.1 (Figure 5—figure supplement 1, Supplementary File 1b) and the experiments were thus performed at a suitable pH. However, as pointed out by the reviewer (and as seen clearly in Figure 3F,G) even small changes in pH can have a dramatic effect on the PcTx1-induced ΔF. It is thus possible that part of the reason why the signal is reduced in 5C is because of small variations in pH. Importantly, we also examine the stability of the ECD_only_ state that is not necessarily dependent on the initial signal size (e.g. Figure 3G, see comment above). So, while we acknowledge that small pH variations might account for the reduced signal, we did our best to correct for it and base our conclusions not only on signal size, but also on the inferred stability of the ECD_only_ state.

Ultimately, I think this is a very interesting and useful paper which will make a sizeable impact in the field and completely solving the mechanism of the PcTx1-induced δ F shouldn't be a requirement for publication.Reviewer #3 (Recommendations for the authors):1. The main claim in this study is that changes in the magnitude of fluorescent signals generated by fluorescent molecules attached to ASIC1a arise from a global conformational change of the ECD induced by binding of PcTx1 to the channel. Since the most important evidence relies on the interpretation of this measurement, I would like further explanation on the following points:a) The two residues chosen for placing the fluorescent labels (V80 and K105) are entirely exposed to water (resting, open, desensitized) in all the structures of human and chicken ASIC available today. Therefore, it is expected that the environment of those sites would not change significantly. Can the authors explain the reason of choosing specifically those two residues?

It is correct that the chosen residues show little displacements in structures of different channel states. Yet, a large number of residues throughout the ASIC ECD have been shown to be successfully labeled for VCF experiments. This has led to new knowledge about the coordinated movements of different domains during gating by for example the Kellenberger group (PMID: 24344244, PMID: 28320963). The reason for being able to detect fluorescence changes despite the absence of large conformational changes uncovered by structural methods is likely twofold: First, the fluorophores are exquisitely sensitive to even small changes in the chemical environment and VCF can thus detect even small and/or transient changes in the orientation of even a single side chain (which may not be apparent from structural work). Second, structures are only capable of providing snapshot of low-energy conformations of the channel and it is not necessarily known if they are functionally relevant. It is therefore possible (even likely!) that the number of possible conformations of ASIC1a exceeds the number of currently known structures. We have now made it clearer in the text on page 4 that we work with Cys mutations that have already been established previously and have been used extensively by others for VCF work. Author response image 1, we are also including a control experiment where we exposed WT channels to the fluorescent dye and conduct the same recording as with K105C. The small change in fluorescence observed here is likely due to perfusion altering the position of the oocyte and is in fact absent in most control recordings with WT channels (see also previous work by us (PMID: 28949138) and others (PMID: 24344244, PMID: 28320963)).

**Author response image 1. sa2fig1:** current (upper trace in black) and fluorescence (lower trace in red) recorded from labeled oocytes expressing K105C or WT mASIC1a.

b) What is the mechanism of the small decrease in fluorescence seen after PcTx1 and pH 5.5 in Figure 1A and Figure 2A. Activation prior to the addition of PcTx1 produces a large increase in fluorescence in channels with K105C* but produces a small negative deflection when channels are activated after addition of the toxin; i.e., is the fluorescent signal produced only by desensitization but activation is silent? Is the downward deflection an artifact or reflects a physical process dismissed in the study? Please, provide an explanation.

Thank you for bringing up this point that we indeed did not comment on so far. The downward deflection is typically overserved in response to a strongly activating proton concentration, e.g. pH 5.5. In most recordings with K105C, the fluorescence signal of the pH 5.5 stimulus first reaches a peak before decreasing slightly towards a plateau. The decrease to this plateau level is what can be seen as negative deflections in pH 5.5 after PcTx1 exposure. We hypothesize that the peak likely reflects a short-lived open or other transient state, while the plateau state is likely related to a desensitized state (see also Figure S1A). This hints at the PcTx1-induced ECD_only_ state indeed being different from e.g. a SSD state. It is not uncommon to see such multi-phasic fluorescent signals in ASICS (PMID: 24344244, PMID: 28320963), but the exact nature of them is often difficult to establish. This is especially true for oocyte recordings, which lack sufficiently high temporal resolution to accurately determine the kinetics of current and fluorescence. We have now added a short statement on the topic when describing the results on page 4.

c) Figure 2A the fluorescent signal shows persistence of 'a long-lasting desensitized conformation' induced by PcTx1 but channels are functional not desensitized. Please, explain what factor(s) determines the shift from one mode the other and whether in a single cell there is heterogeneity of modes.

We could not find this exact formulation in the text, but we assume that the reviewer refers to the functional washout versus the washout of the fluorescence of the PcTx1-induced state. In Figure 2A, the presence of PcTx1 induces a change in fluorescence and inhibits subsequent channel activation. It thus has an effect on the ECD conformation and the pore ('Global'). Upon washout, the current amplitude recovers within the first few minutes, indicating that the functional inhibitory effect of PcTx1 is lost. Conversely, the PcTx1-induced fluorescence change decreases only slightly over the course of the 3 min washout and at the end of the washout we are thus in a mode where we still see a conformational effect on the ECD, while the inhibitory effect on the pore is lost (ECD_only_). This mode therefore differs from the initial 'Global' state. As depicted in Figure 2D, one of the main differences between the two modes is the presence or absence of PcTx1 in the buffer for the 'Global' and 'ECD_only_' state, respectively. Reintroduction of PcTx1 to the buffer can thus lead from the 'ECD_only_' back to the 'Global' state, as now show in the new experiments in Figure 2G, H. We have now made changes to the text to describe the modes in more detail on pages 5, 6, 14 and 15 and have added the corresponding binding modes to the VCF traces in Figures 2, and Figure 2—figure supplement

1.

As for the second part of the above comment: In a single cell we would indeed expect there to be a heterogeneity of modes, which we now also address in the discussion on page 15.

d) The protocol in the first trace of Figure 2A shows multiple activations with low pH. Have the authors considered whether channel activation plays a role in regaining function while the fluorescent signal remains high. In this case the protocol will differ by waiting 3 min after PcTx1 to apply the first activation.

This is an interesting question. We have now included this experiment in Figure 2—figure supplement 1A and B and on page 6. The functional recovery from inhibition is not dependent on activation, but rather washout time.

e) The authors write in the first paragraph of results: 'During a 3 min washout out with pH 7.4 solution, the fluorescence signal displayed only a modest decrease, while exposure …' Do they mean that the toxin was washed out during that period? PcTx could remain bound to the channel for much longer than 3 min as shown by Dawson et al., (DOI: 10.1038/ncomms1917) using SPR at pH 6.3 they measured KD = 1.79x10-9 M, Kon=7.99x105 M s, and Koff=1.4x10-3 that is a tao ~12 min, suggesting that after 3 min there is almost no washout of the toxin. The presence or absence of the toxin attached to the channels needs to be verified with additional experiments. Another caveat, is that affinity of the toxin may be also be pH dependent, complicating the interpretation of changes in fluorescence attributed to pH.

This is a crucial point. While Dawson *et al.,* could measure the k_on_ and K_off_, the measurements do not offer any insight on the functional effect that PcTx1 has on the channel after a washout. In our work, we can actually show that the toxin leads to long-lasting conformational changes in the ECD, while it loses the ability to functionally inhibit the pore (Figure 2A). We have thus provided additional information on the presence of different PcTx1 binding modes that differentiate between the functional inhibitory effect on the pore and the conformational effects on the ECD. To address this point explicitly, we now bring up Dawson *et al.,* in this context in the discussion on page 15.

What our method also shows is indeed a dependence on pH. Changes between the 'Loose' and the 'ECD_only_' state appear to be highly pH dependent (e.g. Figure 1E–H, 2B). It also seems that the wash-off of the ECD_only_ mode is increased at high pH (Figure 1F, H).

2. The MN focuses in a narrow and highly specialized topic requiring a great deal of familiarity with the subject. The Introduction could make a better job in this regard. The following sentences in the Abstract are unintelligible prior to reading the whole MN: 'The loose and ECDonly modes are decoupled from the pore, with the latter inducing a long-lived ECD conformation that reduces the acivity of an endogenous neuropeptide. This long-lived conformational state is proton-dependent and can be destabilized by the F350L mutation". The MN will benefit from deeper description of modes, how are generated, and what underlies the long lived 'desensitization' while regaining response to protons.

We regret that our initial manuscript included language that made is difficult to follow the narrative and thank the reviewer for pointing this out. We have now rephrased parts of the abstract and introduction and describe the different binding modes in greater detail in figures and text and hope to have increased the readability for a broader audience.

3. It would be informative to add to Figure 1B and 1C the fluorescence traces below the currents of activation and SSD. Here, the concentration of PcTx1 is 30 nM, ten-fold lower than in subsequent experiments (300 nM) but is sufficient to shift to the right the pH50ssd and pH50a by one pH unit. The question is whether the magnitude of the fluorescence signal is smaller or remains as in the experiments done with 300 nM. Also, if the fluorescent signal persists or wanes with subsequent activations.

The experiments in Figure 1B–D are electrophysiology experiments conducted on an unlabeled WT mASIC1a construct and thus cannot report on changes in fluorescence signal. The WT and fluorescently labelled constructs have intrinsically different PcTx1 sensitivities, and we thus characterize them separately and refrain from making direct comparisons between these.

Author response image 2, we show that application of 30 nM PcTx1 leads to a slow increase in the fluorescent signal that does not reach a plateau within the 80 s that we typically use when applying PcTx1 (left). Longer application leads to a plateau after around 170 s (middle), with the total fluorescence deflection being comparable to that reached by application of 300 nM PcTx1 for 80s (right). To avoid complications arising from the slow and variable kinetics of the fluorescence signal at 30 nM PcTx1, we used 300 nM PcTx1 throughout.

**Author response image 2. sa2fig2:** current (upper trace in black) and fluorescence (lower trace in red) recorded from labeled oocytes expressing K105C mASIC1a. The dashed line indicates the 80 s mark since the start of the PcTx1 application.

As for the second part of the above comment: In additional experiments, we have established that reapplication of PcTx1 does not lead to an increase of the maximally reached fluorescence level and that subsequent activations have no significant effect on the persistent fluorescence signal of the ECD_only_ state and we hope that the data now shown in Figure 2G, H and Figure 2A sufficiently clarify this point.